

SciPost Phys. Lect.Notes 24 (2021)

# Phase transitions in the early universe

**Mark Hindmarsh[1,2]⋆** 📛**, Marvin Lüben[3]∘** 📛**, Johannes Lumma[4]† and Martin Pauly[4]‡** 📛

**1** Department of Physics and Helsinki Institute of Physics,
PL 64, FI-00014 University of Helsinki, Finland
**2** Department of Physics and Astronomy, University of Sussex,
Brighton BN1 9QH, United Kingdom
**3** Max-Planck-Institut für Physik (Werner-Heisenberg-Institut),
Föhringer Ring 6, 80805 Munich, Germany
**4** Institut für Theoretische Physik, Ruprecht-Karls-Universität Heidelberg,
Philosophenweg 16, 69120 Heidelberg, Germany

⋆ mark.hindmarsh@helsinki.fi, ∘ mlueben@mpp.mpg.de,
†j.lumma@thphys.uni-heidelberg.de, ‡m.pauly@thphys.uni-heidelberg.de

## Abstract

These lecture notes are based on a course given by Mark Hindmarsh at the 24th Saalburg Summer School 2018 and written up by Marvin Lüben, Johannes Lumma and Martin Pauly. The aim is to provide the necessary basics to understand first-order phase transitions in the early universe, to outline how they leave imprints in gravitational waves, and advertise how those gravitational waves could be detected in the future. A first-order phase transition at the electroweak scale is a prediction of many theories beyond the Standard Model, and is also motivated as an ingredient of some theories attempting to provide an explanation for the matter-antimatter asymmetry in our Universe. Starting from bosonic and fermionic statistics, we derive Boltzmann's equation and generalise to a fluid of particles with field dependent mass. We introduce the thermal effective potential for the field in its lowest order approximation, discuss the transition to the Higgs phase in the Standard Model and beyond, and compute the probability for the field to cross a potential barrier. After these preliminaries, we provide a hydrodynamical description of first-order phase transitions as it is appropriate for describing the early Universe. We thereby discuss the key quantities characterising a phase transition, and how they are imprinted in the gravitational wave power spectrum that might be detectable by the space-based gravitational wave detector LISA in the 2030s.

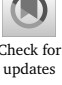

# 1   Introduction

These lecture notes are intended to provide an introduction to the topic of phase transitions in the early universe, focusing on a possible first-order phase transition at temperatures around the scale of electroweak symmetry-breaking, which the universe reached at an age of around $10^{-11}$ s.

Phase transitions are a generic, but not universal, feature of gauge field theories, like the Standard Model, which are based on elementary particle mass generation by spontaneous symmetry-breaking [1,2]. When there is a phase transition in a gauge theory it is (except for special parameter choices) first-order, which means that just below the critical temperature, the universe transitions from a metastable quasi-equilibrium state into a stable equilibrium state, through a process of bubble nucleation, growth, and merger [3–6]. Such a first-order phase transition in the early universe naturally leads to the production of gravitational waves

[7,8]. If it took place around the electroweak scale, by which we mean temperatures in the range 100 – 1000 GeV, the gravitational wave signal could lie in the frequency range of the upcoming space-based gravitational wave detector LISA (Laser Interferometer Space Antenna) [9]. The approval of the mission, and the detection of gravitational waves [10], has generated enormous interest in phase transitions in the early universe.

While the Standard Model has a crossover rather than a true phase transition [11], many extensions of the Standard Model, e.g. with extra scalar fields, lead to first-order phase transitions at the electroweak scale. Gravitational wave signatures are therefore a fascinating new window towards new physics, complementary to that provided by the Large Hadron Collider (see e.g. Ref. [12] for a recent review).

A further motivation for studying electroweak phase transitions is that one of the requirements to explain the matter-antimatter asymmetry in the universe [13] is a departure from thermal equilibrium, which is inevitable in a first-order phase transition. The asymmetry is quantified in terms of the net baryon number of the universe, leading to the name baryogenesis. We will unfortunately not have time to study electroweak baryogenesis in these lectures, and refer the interested reader to e.g. Refs. [14–17].

Let us also briefly mention that non-thermal phenomena might lead to a sizable gravitational wave background. One example of this is the production of gravitational waves associated with preheating; at the end of inflation the inflaton decays to Standard Model particles. The resulting distributions might strongly deviate from local thermal equilibrium. Their violent dynamics could lead to a background of gravitational waves [18–20]. We will not touch on this topic and refer the interested reader to Refs. [17, 21, 22].

A thorough study of early universe phase transitions, gravitational wave production and detection, requires quite a lot of theoretical apparatus from particle physics and cosmology, which could not be covered in a short lecture course. It is assumed that the student has done advanced undergraduate courses on statistical physics and general relativity, and has been introduced to particle physics and cosmology. Wherever possible, the full machinery of thermal quantum field theory is avoided. The aim is to provide a direct route to some important results, and motivate further study and, we hope, research.

The main points we would like the reader to take away are: that the gravitational wave power spectrum from a first-order phase transition is calculable from a few thermodynamic properties of matter at very high temperatures; that these parameters are computable from an underlying quantum field theory; and that these parameters are measurable by LISA. The final point we would like to make is that we can see in outline how the computations, calculations and measurements could be done, but they are far from concrete methods. There is therefore a lot of exciting work to be done in the years leading up to LISA's launch in 2034 to realise the mission's potential scientific reward. These notes are organised as follows. In Sec. 2 we review basic thermodynamics of non-interacting fields and we discuss the different relevant thermodynamic quantities for both fermions and bosons. In Sec. 3 we introduce weak interactions among the fields and derive the thermal Higgs potential. Further, we summarize phase transitions in the Standard Model as well as in models beyond the Standard Model. In Sec. 4 we consider the distribution function of a relativistic fluid and derive the relativistic Boltzmann equation. We generalise the preceding results and study the hydrodynamics of a fluid with a field-dependent mass in Sec. 5 This set-up is analogous to the hydrodynamics with electromagnetic forces, which is governed by the Vlasov equation. In Sec. 6 we study the transition of the Higgs from the false, symmetric phase to the new symmetry-breaking phase and apply the process to the early universe. After these preliminaries, in Sec. 7 we provide a hydrodynamical description of the phase transition in the early unverse. We then discuss the different sources of gravitational waves during a first-order phase transition and the expected power spectra in Sec. 8. Finally, we provide a summary of these lectures and comment on

open issues in Sec. 9.

*Conventions.* Throughout these notes we set $\hbar = k_{\mathrm{B}} = c = 1$ and just re-introduce these constants occasionally. We try to stick to a $(-,+,+,+)$ metric signature. 4-vectors are denoted by roman letters, e.g, $x$, $p$, and $F$ with greek letters as space-time indices, e.g., $\mu, \nu = 0, 1, 2, 3$. Spatial indices are latin letters, e.g., $i, j = 1, 2, 3$ and we denote 3-vectors with an arrow, e.g., $\vec{x}$ and $\vec{p}$.

## 2 Thermodynamics of free fields

We start by studying thermodynamic properties of free bosonic and fermionic fields. For both cases, we derive the partition function, from which all thermodynamic quantities can be derived. We will be particularly interested in the free energy, as it can be used to find the equilibrium states of a theory.

### 2.1 Basic thermodynamics - the bosonic harmonic oscillator

The basic object of thermodynamics is the partition function

$$Z(T) = \mathrm{Tr}\left[e^{-\beta \hat{H}}\right], \tag{2.1}$$

where $\hat{H}$ is the Hamiltonian operator and $\beta = 1/T$ the inverse temperature with $T$ the temperature. The free energy, entropy and energy of the system are given by

$$F = -T \ln Z, \tag{2.2}$$

$$S = -\frac{\partial F}{\partial T}, \tag{2.3}$$

$$E = -\frac{\partial \ln Z}{\partial \beta}, \tag{2.4}$$

respectively. First, we will study a single bosonic harmonic oscillator, the simplest case. To compute its partition function we consider

$$Z_{\mathrm{bho}} = \sum_{n=0}^{\infty} \langle n|e^{-\beta \hat{H}}|n\rangle, \tag{2.5}$$

where the $|n\rangle$ are the eigenstates of the Hamiltonian $\hat{H}$ of the harmonic oscillator with angular frequency $\omega$, together satisfying

$$\hat{H}|n\rangle = \omega \left(n + \frac{1}{2}\right)|n\rangle. \tag{2.6}$$

For the partition function and the free energy this yields

$$Z_{\mathrm{bho}}(T, \omega) = \sum_{n=0}^{\infty} \exp\left[-\beta \omega \left(n + \frac{1}{2}\right)\right] = \frac{e^{-\beta \omega/2}}{1 - e^{-\beta \omega}}, \tag{2.7}$$

$$F_{\mathrm{bho}}(T, \omega) = \frac{1}{2}\omega + T \ln\left(1 - e^{-\beta \omega}\right), \tag{2.8}$$

where the first term in the free energy describes the ground state energy, and the second term is the thermal contribution.

Next we turn to the partition function for a field, or equivalently for a collection of harmonic oscillators. We consider the field operator $\hat{\phi}(\vec{x}, t)$ and decompose it into its Fourier modes

$$\hat{\phi}(\vec{x}, t) = \int \frac{\mathrm{d}^3 k}{(2\pi)^3} \frac{1}{2\omega_{\vec{k}}} \left( \hat{a}_{\vec{k}} e^{-i\vec{k}\cdot\vec{x}} + \hat{a}_{\vec{k}}^{\dagger} e^{i\vec{k}\cdot\vec{x}} \right), \tag{2.9}$$

where we postulate that the operators $\hat{a}, \hat{a}^{\dagger}$ satisfy the commutation relation

$$\left[ \hat{a}_{\vec{k}}, \hat{a}_{\vec{k}'}^{\dagger} \right] = 2\omega_{\vec{k}} (2\pi)^3 \delta^{(3)}(\vec{k} - \vec{k}') \tag{2.10}$$

$$\left[ \hat{a}_{\vec{k}}, \hat{a}_{\vec{k}'} \right] = \left[ \hat{a}_{\vec{k}}^{\dagger}, \hat{a}_{\vec{k}'}^{\dagger} \right] = 0. \tag{2.11}$$

The equation of motion for the free field is the Klein-Gordon equation,

$$\left( \Box + m^2 \right) \hat{\phi}(\vec{x}, t) = 0, \tag{2.12}$$

which in terms of the Fourier modes reads

$$(k^0)^2 = \omega_{\vec{k}}^2 = \vec{k}^2 + m^2. \tag{2.13}$$

This dispersion relation does not involve different momenta and hence the different modes are not coupled. The free scalar field is a collection of independent harmonic oscillators, one for each momentum mode $|\vec{k}|$. The partition function of a bosonic field (indicated by the subscript $B$) thus factorizes into

$$Z_B = \prod_{\vec{k}} Z_{\mathrm{bho}}(T, \omega_{\vec{k}}), \tag{2.14}$$

where the multiplication here is a symbolic notation for a product over all wavenumbers. It can be given meaning by working in finite volume, with the infinite volume limit taken at the end of the calculation.

The free energy of a free bosonic field is given by

$$F_B = -T \ln Z_B = -T \sum_{\vec{k}} \ln Z_{\mathrm{bho}}(T, \omega_{\vec{k}}) \tag{2.15}$$

$$= \sum_{\vec{k}} \left[ \frac{1}{2} \omega_{\vec{k}} + T \ln \left( 1 - e^{-\beta \omega_{\vec{k}}} \right) \right]. \tag{2.16}$$

Again, the sum over all momenta $\vec{k}$ is defined over a finite volume $\mathcal{V}$. In the infinite volume limit $\mathcal{V} \to \infty$, the sum is replaced by an integration as

$$\sum_{\vec{k}} \to \mathcal{V} \int \frac{\mathrm{d}^3 k}{(2\pi)^3}. \tag{2.17}$$

The free energy density, i.e., the free energy normalized to the volume $\mathcal{V}$, then becomes

$$f_B = \frac{F_B}{\mathcal{V}} = V_{0,B} + T \int \frac{\mathrm{d}^3 k}{(2\pi)^3} \ln \left( 1 - e^{-\beta \omega_{\vec{k}}} \right). \tag{2.18}$$

The first term $V_{0,B}$ is the energy density of the zero-temperature ground state, which is divergent, cf. Eq. (2.16). The same divergence is encountered in quantum field theories at zero temperature. In the following, we assume that it is regularized with an appropriate counterterm. The standard renormalisation convention takes the zero-temperature ground state free energy to be zero.

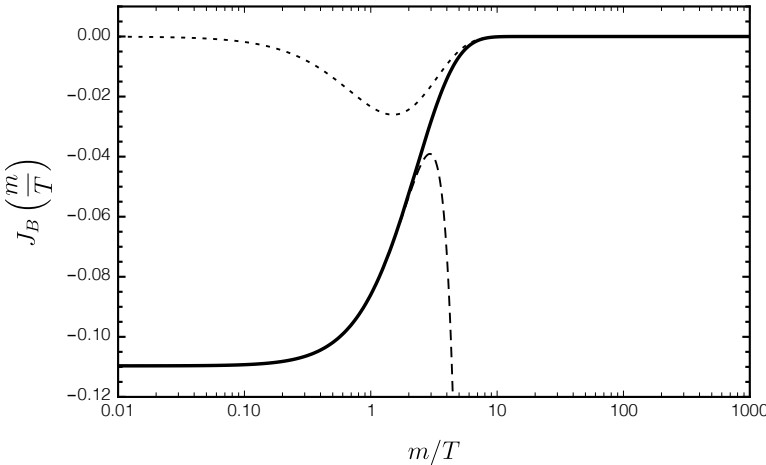

Figure 1: This figure shows the dimensionless function $J_B$ that is proportional to the free energy of bosons as defined in Eq. (2.19), as a function of mass-to-temperature ratio (thick line). Also the expansions for large $T$ (dashed), Eq. (2.21) and small $T$ (dotted), Eq. (2.20) are shown. The large-T expansion is performed up to order four in $m/T$, being a good approximation up to $m/T \sim 1.1$.

Due to the integration over all momenta, $f_B$ can only depend on $T$ and $m$, where $m$ only appears as $m/T$. From dimensional analysis we infer that the free energy density hence takes the form

$$f_B(T, m) = T^4 J_B\left(\frac{m}{T}\right), \tag{2.19}$$

where $J_B(m/T)$ is a dimensionless function. While the integral in Eq. (2.18) cannot be solved exactly for all values of $m/T$, analytic approximations exist in the low and the high temperature regimes.

In the low temperature regime, $m/T \gg 1$, one expands in $T/m$ and gets

$$J_B\left(\frac{m}{T}\right) = -\left(\frac{m}{2\pi T}\right)^{\frac{3}{2}} e^{-m/T}\left(1 + \mathcal{O}\left(\frac{T}{m}\right)\right), \tag{2.20}$$

recovering the familiar distribution function of Maxwell-Boltzmann statistics.

In the high temperature case $m/T \ll 1$ one expands in $m/T$ to obtain

$$
\begin{aligned}
J_B\left(\frac{m}{T}\right) = {} & -\frac{\pi^2}{90} + \frac{1}{24}\left(\frac{m}{T}\right)^2 - \frac{1}{12\pi}\left(\left(\frac{m}{T}\right)^2\right)^{\frac{3}{2}} \\
& -\frac{1}{2(4\pi)^2}\left(\frac{m}{T}\right)^4\left[\ln\left(\frac{1}{4\pi}\frac{m}{T}e^{\gamma_E}\right) - \frac{3}{4}\right] \\
& +\mathcal{O}\left(\left(\frac{m}{T}\right)^6\right).
\end{aligned}
\tag{2.21}
$$

Here $\gamma_E \approx 0.57721$ is the Euler–Mascheroni constant. While the first two terms follow in a relatively simple way using the $\zeta$-function, the third and fourth terms are more complicated in nature: they are non-analytic in the fundamental expansion parameter $m^2/T^2$, and can only be derived using more advanced methods. For details, we refer to Ref. [23].

A numerical evaluation of the function $J_B(m/T)$ is depicted in Fig. 1, along with its high and low temperature approximations. It can be seen that the high temperature approximation is good even up to $m/T \simeq 2$.

## 2.2 The fermionic harmonic oscillator

For fermions the Pauli exclusion principle holds: a quantum state can only be occupied by a single fermion at once. To compute the fermionic partition function, we therefore sum over only the occupation numbers 0 and 1 in order to respect the Pauli exclusion principle, arriving at

$$Z_{\text{fho}}(T, \omega) = \sum_{n=0}^{1} \langle n | e^{-\beta \hat{H}} | n \rangle = e^{\beta \omega / 2} \left( 1 + e^{-\beta \omega / 2} \right), \tag{2.22}$$

after using Eq. (2.6). Consequently, the free energy for fermions is given by

$$F_{\text{fho}}(T, \omega) = -\frac{1}{2} \omega - T \ln \left( 1 + e^{-\beta \omega} \right). \tag{2.23}$$

In order to generalize the above expression to fermionic fields we introduce a Dirac spinor field $\hat{\Psi}_\alpha(\vec{x}, t)$, which creates and destroys massive fermions. Every such spinor has four components, denoted by the index $\alpha$. The components describe particles and antiparticles, each of which have two spin degrees of freedom.

The spinor can be decomposed into Fourier modes, where the Fourier coefficients are operators subject to a set of anticommutation relations, which need to respect the fermionic nature of $\Psi_\alpha$. Analogously to the bosonic case, the free fermionic field is a collection of independent harmonic oscillators, four for each momentum mode $|\vec{k}|$. The partition function for a fermionic field $F$ hence factorizes into

$$Z_F = 4 \prod_{\vec{k}} Z_{\text{fho}}(T, \omega_{\vec{k}}), \tag{2.24}$$

where $Z_{\text{fho}}$ is the partition function of a single fermionic harmonic oscillator, cf. Eq. (2.23). This leads to the fermionic free energy

$$F_F = -4 \sum_{\vec{k}} \left[ \frac{1}{2} \omega_{\vec{k}} + T \ln \left( 1 + e^{-\beta \omega_{\vec{k}}} \right) \right]. \tag{2.25}$$

The factor of 4 arises due to the four components of an individual spinor. In the continuum limit (infinite volume) one needs to replace the sum by an integration, as in Eq. (2.17). The free energy density for each fermionic degree of freedom is thus

$$f_F = -V_{0,F} - T \int \frac{\mathrm{d}^3 k}{(2\pi)^3} \ln \left( 1 + e^{-\beta \omega_{\vec{k}}} \right). \tag{2.26}$$

The vacuum energy density $V_{0,F}$ is again divergent, but comes with the opposite sign compared to the bosonic case. We assume that the vacuum energy is regularised by an appropriate counter-term such that we can take it to be zero in the following.

In analogy to the bosonic case the free energy density can be written as

$$f_F = T^4 J_F \left( \frac{m}{T} \right). \tag{2.27}$$

In Fig. 2, a numerical evaluation of $J_F(m/T)$ is shown together with its small and large temperature expansions. The integral in Eq. (2.26) cannot be solved analytically for all values of $m/T$, but in the limits of small and large temperatures. In the low temperature limit the expansion agrees with Eq. (2.20). As expected, at low energies the quantum nature of the field becomes irrelevant and one recovers the Maxwell-Boltzmann statistics for both, fermions

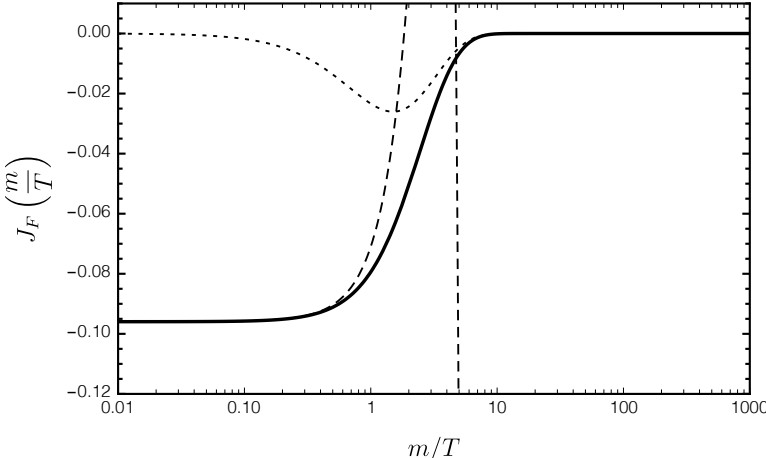

Figure 2: This figure shows the dimensionless function $J_F$ that is proportional to the free energy of fermions as defined in Eq. (2.27) as a function of mass-to-temperature ratio (thick line). Additionally, the expansion for large $T$ (dashed), Eq. (2.28) and small $T$ (dotted) , in analogy to Eq. (2.20) are shown. Note that in the small $T$ limit, both the fermionic and the bosonic expansions agree. The large $T$ expansion is performed up to order four in $m/T$, working well up to $m/T \sim 0.5$, hence being sligthly worse than the bosonic high-T expansion, depicted in Fig. 1 .

and bosons. In the high temperature limit the free energy can be approximated as

$$
\begin{aligned}
J_F\left(\frac{m}{T}\right) \;=\; & -\frac{7}{8}\frac{\pi^2}{90} + \frac{1}{48}\left(\frac{m}{T}\right)^2 \\
& -\frac{1}{2(4\pi)^2}\left(\frac{m}{T}\right)^4\left[\ln\left(\frac{1}{\pi}\frac{m}{T}e^{\gamma_E}\right) - \frac{3}{4}\right] \\
& +\mathcal{O}\left(\left(\frac{m}{T}\right)^6\right).
\end{aligned}
\tag{2.28}
$$

Compared to Eq. (2.21), the term that appeared with a power of 3/2 disappeared, and the first term has a prefactor of 7/8. Eqs. (2.27) and (2.28) together give the free energy of a single Dirac fermion field.

## 3 Phase transition in field theory

After having discussed the free energy of free bosons and fermions, let us introduce interactions among these fields. In a weakly interacting field theory one can compute the free energy as a perturbation around the free energy of a free field. In this section we will present a setup, which is tailored to discuss phase transitions in the Standard Model. Phase transitions in weakly coupled gauge theories were first discussed in Refs. [1, 2].

### 3.1 The Standard Model at weak coupling

In the Standard Model the masses of fermions and gauge bosons $M_i$ depend linearly on the magnitude of the Higgs field $\phi$,

$$
M_i(\phi) = c_i \phi\,,
\tag{3.1}
$$

with the $c_i$ proportional to the dimensionless coupling constants. The index $i$ labels the Standard Model fields that couple to the Higgs. Today, the Higgs is in its broken phase and the

Table 1: The zero-temperature masses and the mass proportionality constants $c_i$, defined in Eq. (3.1), for the most massive fields in the Standard Model. Here, $y_t$ is the Higgs-Yukawa coupling of the top quark, $\lambda$ the Higgs self-coupling, and $g$ and $g'$ the gauge couplings of the $W_\mu^a$ and $B_\mu$ bosons.

| particle | mass [GeV] | $c_i$ |
|:---:|:---:|:---:|
| $t$ | 172.76 | $y_t$ |
| $H$ | 125.18 | $\sqrt{2\lambda}$ |
| $Z$ | 91.19 | $\sqrt{g^2 + g'^2}/\sqrt{2}$ |
| $W^\pm$ | 80.38 | $g/\sqrt{2}$ |

Table 2: An overview over events happening at different energy scales in the early universe. These determine the effective number of degrees of freedom in the Standard Model at a certain energy scale.

| energy scale | event |
|---:|:---|
| 100 GeV | $t$ non-relativistic |
| 1 GeV | $b$ non-relativistic |
| 500 MeV | $c$, $\tau$ non-relativistic |
| 200 MeV | QCD phase transition |
| 30 MeV | $\mu$ non-relativistic |
| 2 MeV | $\nu$ freeze-out |
| 0.2 MeV | $e$ non-relativistic |
| 1 eV | matter-radiation equality |
| 0.1 eV | photon decoupling |

Higgs field assumes its vacuum expectation value, $\phi = v_{EW} \simeq 246\,\text{GeV}$, which determines the particle masses we observe in experiments. This value for the field is dynamically determined by the minimisation of the zero-temperature potential for the Higgs field,

$$V_0(\phi) = \frac{\lambda}{4} \left( \phi^2 - v_{EW}^2 \right)^2 . \tag{3.2}$$

The relation of the $c_i$ to the standard coupling constants of the Standard Model are given for the most massive fields in Tab. 1. One can see that for these fields, the values of $c_i$ are all $\mathcal{O}(1)$. For the other fields of the standard model, $c_i \ll 1$.

The Higgs particle is a quantised fluctuation around the ground state, with mass

$$M_H = \sqrt{V_0''(v_{EW})} = \sqrt{2\lambda} v_{EW}.$$

The Higgs field is unique in that its mass does not in general depend linearly on $\phi$. At this level of treatment, we will not need to know that in the Standard Model, the Higgs field is a two-component vector of complex scalar fields $\Phi$, but for completeness we mention that $\phi^2 = \Phi^\dagger \Phi/2$.

The free energy density $f$ of a gas of Standard Model particles is given by the zero temperature result (3.2), plus terms that arise due to the interaction with the Higgs according to Eq. (3.1). This gives

$$f = V_0(\phi) + \sum_B f_B + \sum_F f_F \tag{3.3}$$

$$= V_0(\phi) + T^4 \left[ \sum_B J_B \left( \frac{M_B}{T} \right) + \sum_F J_F \left( \frac{M_F}{T} \right) \right]. \tag{3.4}$$

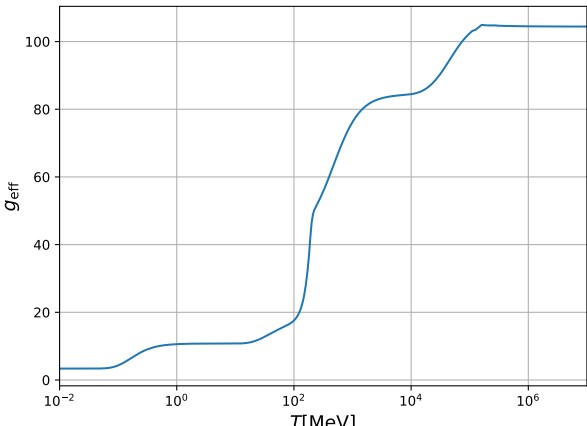

Figure 3: This figure shows the effective number of relativistic degrees of freedom $g_{\text{eff}}$ of a Standard Model plasma as a function of temperature, taking into account interactions between particles, with both perturbative and lattice methods [24].

Here, we sum over all fermions $F$ and bosons $B$ that are relativistic at temperature $T$. For large temperatures, we can write the free energy density as

$$f = -g_{\text{eff}}\frac{\pi^2}{90}T^4 + V_T(\phi). \tag{3.5}$$

Here, $g_{\text{eff}}$ is the effective number of relativistic degrees of freedom, given by

$$g_{\text{eff}} = \frac{7}{8}4N_{\text{F}} + 3N_{\text{V}} + 2N_{\text{V0}} + N_{\text{S}}, \tag{3.6}$$

where $N_{\text{F}}$ is the number of Dirac fermions, $N_{\text{V}}$ is the number of massive vectors, $N_{\text{V0}}$ is the number of massless vectors and $N_{\text{S}}$ is the number of scalars. The prefactors account for the degrees of freedom of each of the particles. In the case of only bosons or only fermions this expression reduces to the first term in Eq. (2.21) or Eq. (2.28), resp. For the Standard Model at high energies this value is $g_{\text{eff}} = 106.75$.[1] As the temperature decreases, so does the effective number of relativistic degrees of freedom, as more and more particles become non-relativistic, or are bound together into hadrons. The function $g_{\text{eff}}(T)$ for the Standard Model is shown in Fig. 3, using data taken from Ref. [24], where interactions between particles (and not just the mass generation effect of the Higgs) are also taken into account. Tab. 2 summarizes key temperatures that affect $g_{\text{eff}}$.

The second, field-dependent term in Eq. (3.5), $V_T(\phi)$, is called the thermal effective Higgs potential. According to Eqs. (2.21) and (2.28) it is given by [23,25–27]

$$\begin{aligned}
V_T(\phi) &= V_0(\phi) + \frac{T^2}{24}\left(\sum_S M_S^2(\phi) + 3\sum_V M_V^2(\phi) + 2\sum_F M_F^2(\phi)\right) \\
&\quad - \frac{T}{12\pi}\left(\sum_S \left(M_S^2(\phi)\right)^{\frac{3}{2}} + \sum_V \left(M_V^2(\phi)\right)^{\frac{3}{2}}\right) \\
&\quad + \text{higher order terms}.
\end{aligned} \tag{3.7}$$

Here $M_S$, $M_V$, $M_F$ are the masses of the scalar fields $S$, vector fields $V$ and fermionic fields $F$, which are related to the expectation value of the Higgs as in Eq. (3.1).

---

[1]It is a good exercise to verify this. Note that the neutrinos of the Standard Model count as $N_F = 1/2$, as they are two-component spinors.

For small field values the expression for the scalar masses in Eqn. (3.7) is negative, leading to imaginary contributions to the thermal Higgs potential. This problem does not arise if instead of the bare mass one takes into account the thermal mass in (3.7). However, the thermal mass itself depends on the thermal potential. A self-consistent solution then requires resummation techniques [28,29]. In addition, technical subtleties associated with gauge choice in the Higgs sector arise [30] - the resulting effective potential is gauge dependent, while observable quantities remain gauge independent, as they should. For further details we refer the reader to Refs. [27,31].

For high temperatures, the thermal effective potential can be approximated by an expansion in $\phi/T$ yielding

$$V_T(\phi) = \frac{D}{2}\left(T^2 - T_0^2\right)\phi^2 - \frac{A}{3}T\phi^3 + \frac{\lambda_T}{4!}\phi^4 + \dots, \tag{3.8}$$

where $A$, $D$ are constants and $\lambda_T$ depends only logarithmically on the temperature. In the Standard Model we have

$$A = \frac{1}{12\pi\phi^3}\left(M_H^3 + 6M_W^3 + 3M_Z^3\right) \tag{3.9}$$

$$D = \frac{1}{12\phi^2}\left(M_H^2 + 6M_W^2 + 3M_Z^2 + 6M_t^2\right) \tag{3.10}$$

$$\lambda_T \simeq \lambda \tag{3.11}$$

$$T_0 = \sqrt{\frac{1}{2D}}M_H, \tag{3.12}$$

where we have dropped the logarithmic dependence of $\lambda_T$ on $T$. Here, the subscripts $H$, $W$, $Z$, and $t$ denote the Higgs-boson, $W$- and $Z$-bosons, and the top quark of the Standard Model. Notice that only bosons contribute to the cubic term in the potential.

The form of this quartic potential is sketched for different values of the temperature in Fig. 4. For large temperatures, $T \gg T_c$, the potential has a minimum at $\phi = 0$, which is the only ground state or equilibrium state of the system. We will refer to the ground state where $\phi = 0$ also as symmetric phase. As the temperature drops, a second, but higher lying minimum develops as represented by the dark green line. Both minima are degenerate at the critical temperature

$$T_c = T_0\left(1 - \frac{2}{9}\frac{A^2}{\lambda D}\right)^{-\frac{1}{2}}, \tag{3.13}$$

which is well-defined only if $2A^2 < 9\lambda D$. This case will be of particular interest to us, as in this case the two minima at $\phi = 0$ and $\phi(T_c) = 2AT_c/3\lambda$ are separated by a free energy barrier, signaling a first-order phase transition. Below the critical temperature, the system can supercool, staying in the false ground state at $\phi = 0$ for some time, before transitioning to the global minimum. In the bosonic case, this is reflected by the cubic term. We will refer to the ground state where the Higgs field is non-zero also as Higgs phase. For $T = 0$ the thermal corrections are absent and the minimum is at $\phi = v_{EW} \simeq 246\,\text{GeV}$.

## 3.2 Breakdown of weak coupling

So far we have assumed that the free energy is only slightly altered by interactions, an assumption of weak (i.e. small) couplings between particles. Moreover, we only included interactions with the Higgs field, in their simplest form of a mass generation effect. To properly include interactions, one really has to study and apply thermal field theory [27]. In this section we merely sketch where the assumption of weak coupling breaks down, with a qualitative argument using the statistical mechanics of a field in 3 dimensions.

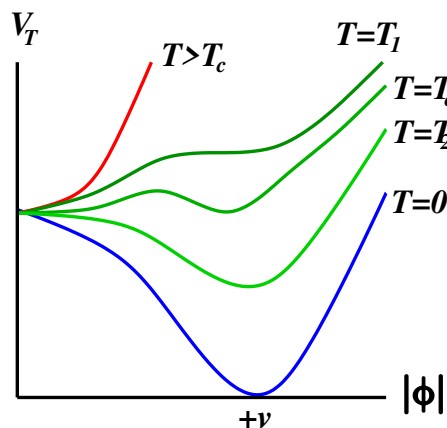

Figure 4: The figure shows the thermal effective Higgs potential $V_T(\phi)$ at different temperatures. For large temperatures $T \gg T_c$ (red) the potential has a minimum at $\phi = 0$ and the ground state is symmetric. Below the temperature $T_1 > T_c$ (dark green) a second, but higher lying minimum develops. At the critical temperature $T_c$ (green) both minima are degenerate. Below the critical temperature, the new minimum at non-zero field value is the global minimum representing the true (stable) ground state.

In an interacting theory, one splits the Hamiltonian $\hat{H} = \hat{H}_0 + \hat{H}_I$ into a free and an interacting Hamiltonian. As an example inspired by the Standard Model consider a scalar field (not necessarily the Higgs) with an interaction Hamiltonian

$$\hat{H}_I = g^2 \int \mathrm{d}^3 x \, \hat{\phi}^4, \tag{3.14}$$

with $g^2$ an arbitrary dimensionless coupling constant. We leave the mass of this scalar field free. Weak coupling means that $g^2 \ll 1$.

We can then try to compute the partition function

$$Z = \mathrm{Tr}\left[ e^{-\beta(\hat{H}_0 + \hat{H}_I)} \right], \tag{3.15}$$

by expanding in powers of the coupling constant. This is a non-trivial exercise, but it turns out that we are in fact expanding in the parameter

$$\varepsilon = g^2 f(\vec{k}), \tag{3.16}$$

with $f$ the phase space density. For a boson,

$$f(\vec{k}) = \frac{1}{e^{\beta \omega_{\vec{k}}} - 1}, \tag{3.17}$$

which approaches $T/\omega_{\vec{k}}$ for frequencies low compared with the temperature, $\omega_{\vec{k}} \ll T$. In this limit, the expansion parameter reads

$$\varepsilon = \frac{g^2 T}{\omega_{\vec{k}}}, \tag{3.18}$$

which is greater than unity for $k \lesssim g^2 T$. The expansion parameter diverges as $|\vec{k}| \to 0$ (in the "infrared") for massless bosons, cf. Eq. (2.13). We therefore learn that in the case of massless bosons at zero chemical potential a perturbative expansion in powers of $g$ breaks down in a thermal state, at any temperature [32], for momenta $k \lesssim g^2 T$.

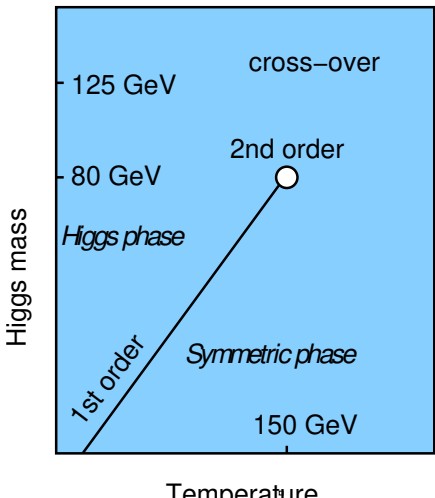

Figure 5: The phase diagram of the Standard model. For Higgs masses of $m_H \lesssim 75$ GeV the Standard Model undergoes a first-order phase transition. For larger Higgs masses, there is no phase transition between the symmetric phase $\phi = 0$ and the Higgs phase $\phi = v_{\text{EW}}$, but a cross-over. Including higher-order interactions changes the picture significantly.

However, thermal corrections contribute to the mass of a thermal state which have to be taken into account. One can apply the above argument to the $W$, $Z$ and gluons of the Standard Model, which have an interaction term of a similar form.[2]

For gauge fields, one distinguishes between the *electric mass* (a mass parameter appearing in the wave equation for the timelike component of the gauge field $A_0$) and the corresponding *magnetic mass* for the spacelike components $A_i$. The timelike component of the gauge field behaves like a scalar field, both of which have a mass-temperature relation as

$$m^2(T) = m_0^2 + c g^2 T^2, \tag{3.19}$$

where $c$ is a theory-dependent constant, and $m_0$ is the mass of the field at zero temperature. Therefore, the expansion parameter $\epsilon$ for massless gauge bosons (such as the photon) with $m_0 = 0$ is of the order of the coupling constant, $\epsilon \sim g \ll 1$. This means that for fields with electric mass perturbation theory is trustworthy at any temperature for small coupling constants. Physically, the electric mass makes the electric field at a distance $r$ from a static charge behave as $r^{-1} \exp[-m(T)r]$, that is, the electric field is screened. The electric mass is none other than the inverse Debye length: the free charges in the plasma become polarised around a source.

The magnetic mass, on the other hand, vanishes in perturbation theory. Therefore, the expansion parameter $\epsilon$ is divergent in the IR and one should be suspicious of perturbation theory. The vanishing of the perturbative magnetic mass turns out not to matter for the photon, as it has no self-interaction terms in its Hamiltonian, but for the other gauge bosons of the Standard Model our naive perturbation theory definitely breaks down. To study phase transitions, more involved methods such as a combination of advanced resummation techniques and lattice simulations are required.

### 3.3 Beyond weak coupling and the Standard Model

In more advanced calculations based on numerical computations of the partition function, the following picture emerges [11, 33–37]. One can study the Standard Model (in fact any gauge theory with spontaneous symmetry-breaking) in the 2-dimensional space spanned by the temperature and the ratio of the Higgs mass to the gauge boson mass. To simplify matters when discussing the Standard Model, one can take the gauge boson mass to be the one of the $W$-boson, $m_W \simeq 80$ GeV, and take one of the parameters to be the Higgs mass. This leads to the picture presented in Fig. 5.

If the ratio of the Higgs mass to the gauge boson mass is small, the simple picture outlined in the previous sections is correct: the perturbative evaluation of the thermal potential is reasonably accurate, and there is indeed a first-order phase transition. This would correspond to the case of the Standard Model if the Higgs mass were much less than 80 GeV. However, as the ratio increases, the strength of the transition, as measured for example by the latent heat, decreases. At a critical value for this ratio, the latent heat goes to zero. Above this critical value the transition is a cross-over.

In case of a cross-over the system smoothly changes from the symmetric phase to the Higgs phase. The situation is similar to water at high pressure, whose density smoothly decreases with temperature, rather than making a sharp transition from the vapour to the liquid phase.

Precisely at the critical ratio, the transition is second-order, meaning that the effective thermal mass of the Higgs $M_H(T)$ goes to zero, and its correlation length diverges. The same phenomenon happens with water at its critical point (374° C and 218 atm), where the divergence of the correlation length can be observed as the phenomenon of critical opalescence. In terms of the zero-temperature Higgs mass, the critical point is at around 80 GeV. Given the measured value for the Higgs mass of 125 GeV, the Standard Model is well into the cross-over region.

However, in theories beyond the Standard Model, the electroweak phase transition can be a first-order phase transition. Indeed already the inclusion of a $\phi^6$ operator in the Higgs potential could lead to a first-order phase transition [38–40]. Such a term is not allowed by the Standard Model, but it could be part of an effective field theory describing new physics.

The motivation to study models with new physics includes providing an explanation for the matter-antimatter asymmetry in the Universe [41] to explaining dark matter [42, 43]. There are further shortcomings of the Standard Model that need to be addressed [43].

Many such extensions of the Standard Model include extra scalars, which can give first-order phase transitions. Examples include coupling the Standard Model to an extra Higgs SU(2) singlet, doublet ("2HDM") or triplet. Further, extensions of the Standard Model with supersymmetry automatically include extra scalars, although it seems that the simplest such extensions do not have a first-order phase transition. Possibilities exist beyond the framework of weakly-coupled field theory. A review of Standard Model extensions with first-order phase transitions is given in a recent LISA Cosmology Working Group report [12].

In the following chapters we will study the details of the dynamics of such a first-order phase transition.

## 4 Relativistic hydrodynamics

In order to understand the dynamics of a first-order phase transition, we need an appropriate hydrodynamic description of the early universe. In this section we study interacting bosonic and fermionic particles in the thermodynamic limit. For the resulting distribution function of

---

[2]Indeed, this infrared problem was first pointed out for gauge bosons [32].

a relativistic fluid, we will derive the relativistic Boltzmann equation. For now we focus on the case where the particle masses are constant throughout spacetime. We mostly follow Ref. [44] in this section.

## 4.1 Distribution function

In the presence of several interacting harmonic oscillators, we need to also include a number operator $\hat{N}$ in the definition of the partition function

$$Z = \text{Tr}\left[e^{-\beta(\hat{H} - \mu\hat{N})}\right], \tag{4.1}$$

where $\mu$ is the chemical potential. The number operator is obtained as

$$\hat{N} = T\frac{\partial}{\partial\mu}\ln Z. \tag{4.2}$$

With the same procedure as described for the 1-particle partition function in Sec. 2, we arrive at the partition functions and number operators for free bosonic, indicated by $B$, and fermionic, indicated by $F$, fields with 3-momentum $\vec{p}$:

$$\text{bosons:} \quad Z_B = \frac{e^{-\beta E_{\vec{p}}/2}}{1 - e^{-\beta(E_{\vec{p}} - \mu)}} \implies N_B = \frac{1}{e^{\beta(E_{\vec{p}} - \mu)} - 1}, \tag{4.3}$$

$$\text{fermions:} \quad Z_F = \frac{e^{\beta E_{\vec{p}}/2}}{1 + e^{-\beta(E_{\vec{p}} - \mu)}} \implies N_B = \frac{1}{e^{\beta(E_{\vec{p}} - \mu)} + 1}. \tag{4.4}$$

From now on we switch to the notation $\omega_{\vec{p}} \to E_{\vec{p}}$ such that the dispersion relation reads $E_{\vec{p}}^2 = \vec{p}^2 + m^2$. The particle number densities are

$$n_B = \frac{N_B}{\mathcal{V}} = \int \frac{\mathrm{d}^3p}{(2\pi)^3}\frac{1}{e^{\beta(E_{\vec{p}} - \mu)} - 1}, \quad n_F = \int \frac{\mathrm{d}^3p}{(2\pi)^3}\frac{1}{e^{\beta(E_{\vec{p}} - \mu)} + 1}. \tag{4.5}$$

Hence, let us introduce the 1-particle distribution function as

$$f_\eta(\vec{p}) = \frac{1}{e^{\beta(E_{\vec{p}} - \mu)} - \eta}, \tag{4.6}$$

where $\eta = +1$ for bosonic fields and $\eta = -1$ for fermionic fields. The $\mu = 0$ case will be the relevant case for us; although the total particle density is high in the early universe, the net particle number densities are very small.

Our aim is to allow small departures from equilibrium, which can be described by local changes in the distribution function, such that it becomes space and time dependent. We will then obtain the evolution equation for the distribution function $f(p,x)\mathrm{d}^3p\,\mathrm{d}^3x$ where $f(p,x)$ describes the average number of particles of momentum $\vec{p}$ in a 3-phase space volume element $(\vec{x}, \vec{x} + \mathrm{d}\vec{x}) \times (\vec{p}, \vec{p} + \mathrm{d}\vec{p})$ at time $t = x^0$. From this function, we can define various quantities like the number density $n(x)$ and the particle flux $j^i(x)$ as

$$n(x) = \int \frac{\mathrm{d}^3p}{(2\pi)^3}f(p,x), \tag{4.7}$$

$$j^i(x) = \int \frac{\mathrm{d}^3p}{(2\pi)^3}\frac{p^i}{p^0}f(p,x). \tag{4.8}$$

We can combine both quantities and define the particle current

$$j^\mu(x) = \int \frac{\mathrm{d}^3p}{(2\pi)^3}\frac{p^\mu}{p^0}f(p,x). \tag{4.9}$$

Furthermore, we define the energy density $e(x)$, the 3-momentum density $\Pi^i(x)$ and the 3-momentum flux $\Pi^{ij}(x)$ as

$$e(x) = \int \frac{\mathrm{d}^3 p}{(2\pi)^3} p^0 f(p, x), \tag{4.10}$$

$$\Pi^i(x) = \int \frac{\mathrm{d}^3 p}{(2\pi)^3} p^i f(p, x), \tag{4.11}$$

$$\Pi^{ij}(x) = \int \frac{\mathrm{d}^3 p}{(2\pi)^3} p^i \frac{p^j}{p^0} f(p, x). \tag{4.12}$$

We can combine the last three quantities to form the energy-momentum tensor,

$$T^{\mu\nu}(x) = \int \frac{\mathrm{d}^3 p}{(2\pi)^3} \frac{p^\mu p^\nu}{p^0} f(p, x), \tag{4.13}$$

where $T^{00} = e$, $T^{0i} = \Pi^i = T^{i0}$, and $T^{ij} = \Pi^{ij}$. The integral measure transforms as a scalar under Lorentz transformations because we can rewrite it as

$$\int \frac{\mathrm{d}^3 p}{(2\pi)^3} \frac{1}{p^0} = \int \frac{\mathrm{d}^4 p}{(2\pi)^4} \delta(p^2 + m^2)\theta(p^0), \tag{4.14}$$

where $\theta(p^0)$ ensures positivity of the energy. Hence, for the energy-momentum tensor $T^{\mu\nu}$ to transform as a 2-tensor under Lorentz-transformation (and the particle current $j^\mu$ as a 1-tensor), the distribution function $f(p, x)$ has to be a Lorentz-scalar.

## 4.2 Relativistic Boltzmann equation

Let us study how the particle distribution function changes in time. First, let us assume that there are no collisions between the individual particles, and follow the trajectory of each particle in phase space $(x^\mu(\tau), p^\mu(\tau))$, which is parametrised by proper time $\tau$. The position and momentum after a small proper time interval $\mathrm{d}\tau$ hence change as

$$x^\mu(\tau) \longrightarrow x^\mu(\tau) + \frac{\mathrm{d}x^\mu}{\mathrm{d}\tau}\mathrm{d}\tau = x^\mu(\tau) + \frac{p^\mu}{m}\mathrm{d}\tau, \tag{4.15}$$

$$p^\mu(\tau) \longrightarrow p^\mu(\tau) + F^\mu \mathrm{d}\tau, \tag{4.16}$$

where $F^\mu$ describes an external 4-force. This is sketched in Fig. 6, which shows the phase space at two time steps $\tau$ and $\tau + \mathrm{d}\tau$. The particle distribution functions at time $\tau$ and $\tau + \mathrm{d}\tau$ must be the same because the particle number is conserved in an infinitesimal phase space volume. This leads to

$$f\left(p + F\mathrm{d}\tau, x + \frac{p}{m}\mathrm{d}\tau\right) = f(p, x). \tag{4.17}$$

A Taylor expansion around $\mathrm{d}\tau = 0$ yields

$$\left(p^\mu \partial_\mu + mF^\mu \frac{\partial}{\partial p^\mu}\right) f(p, x) = 0. \tag{4.18}$$

A consistent differential equation for the distribution function needs to maintain the on-shell condition $p^2 + m^2 = 0$. This is the case when the force satisfies either (a) $F^\mu p_\mu = 0$ or (b) $F^\mu = -\partial^\mu m$. We introduce the on-shell condition by hand to arrive at the *collisionless relativistic Boltzmann equation*

$$\left(p^\mu \partial_\mu + mF^\mu \frac{\partial}{\partial p^\mu}\right) \delta(p^2 + m^2) f(p, x) = 0. \tag{4.19}$$

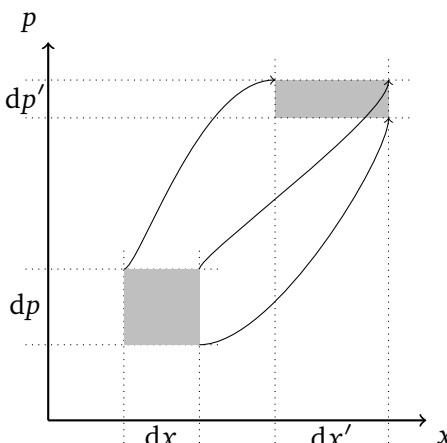

Figure 6: This figure depicts how the phase space volume occupied by particles within $(x, x+\mathrm{d}x)$ and $(p, p+\mathrm{d}p)$ changes in a time step $\mathrm{d}\tau$ to $(x', x'+\mathrm{d}x')$ and $(p', p'+\mathrm{d}p')$. The particle number in both elements is the same.

Next, we want to incorporate collisions between the particles. We focus on 2-body collisions between classical particles only, as these dominate the scattering in the energy regime that we are interested in. Furthermore, we assume that there are no external forces, $F^\mu = 0$. Let us denote initial values, i.e., before the collision, without a prime and final values with a prime (see Figure 7). Momentum conservation requires

$$p_1 + p_2 = p_1' + p_2' , \tag{4.20}$$

where the subscript refers to the particle. In the presence of collisions, the particle distribution function is not necessarily the same after a time step $\mathrm{d}\tau$ because scattering can remove and add particles to the phase space volume element $(\mathrm{d}p, \mathrm{d}x)$ around $(p, x)$. Let $R$ ($\bar{R}$) describe a scattering in time $\mathrm{d}t$ which removes (adds) an initial (final) particle with momentum $p$ at position $x$. The Boltzmann equation including collisions reads

$$p^\mu \partial_\mu f(p, x) = \bar{R}(p, x) - R(p, x) . \tag{4.21}$$

Let us quantify $R$ and $\bar{R}$ a bit further. An incoming particle with momentum $p_1$ can scatter with any particle that is at the same position, but with arbitrary momentum $p_2$ (and likewise for outgoing). That amounts to writing

$$R(p_1, x) = \int \frac{\bar{\mathrm{d}}^3 p_2}{2E_2} \frac{\bar{\mathrm{d}}^3 p_1'}{2E_1'} \frac{\bar{\mathrm{d}}^3 p_2'}{2E_2'} f(p_1, x) f(p_2, x) W(p_1, p_2 | p_1', p_2') \tag{4.22}$$

$$\bar{R}(p_1, x) = \int \frac{\bar{\mathrm{d}}^3 p_2}{2E_2} \frac{\bar{\mathrm{d}}^3 p_1'}{2E_1'} \frac{\bar{\mathrm{d}}^3 p_2'}{2E_2'} f(p_1', x) f(p_2', x) W(p_1', p_2' | p_1, p_2) , \tag{4.23}$$

where $W$ is called the scattering function. Here, we introduced the short-hand notation $\bar{\mathrm{d}}^3 p = \mathrm{d}^3 p / (2\pi)^3$. We can write $W$ in terms of the cross section $\sigma$ as

$$W(p_1, p_2 | p_1', p_2') = s\, \sigma(s, \theta) \delta^{(4)}(p_1' + p_2' - p_1 - p_2) \tag{4.24}$$

in terms of the Mandelstam variables $s = (p_1 + p_2)^2$, $t = (p_1 - p_1')^2$, and the scattering angle $\theta$ is obtained from $\cos \theta = 1 - 2t/(s + 4m^2)$.

Let us introduce the *collision function* $C[f] = \bar{R} - R$, where the notation reminds us that $R$ and $\bar{R}$ depend on the distribution function. We now have derived the *relativistic Boltzmann equation*

$$p^\mu \partial_\mu f = C[f] . \tag{4.25}$$

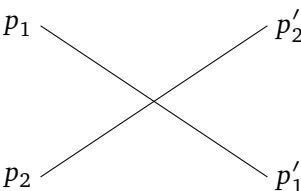

Figure 7: Visual depiction of a 2-body scattering event. The unprimed quantities describe the values before the scattering, the primed ones after the scattering.

## 4.3 Conservation laws and collision invariants

It is natural to expect that conservation laws obeyed during the collisions constrain the evolution of the distribution function. In our above system with 2-body collisions, we expect that the particle number is conserved, as well as the momentum. One can show that for any function $\psi(p,x) = a(x) + b_\mu(x)p^\mu$ with $a$ and $b_\mu$ arbitrary functions of $x$, the following integral vanishes identically,

$$\int \frac{\bar{\mathrm{d}}^3 p}{2E}\psi(p,x)C[f] = 0\,, \tag{4.26}$$

as a consequence of particle number and momentum conservation. The function $\psi$ is a *collision invariant*.

Now we replace the collision function using the Boltzmann equation (4.25). For the case $b_\mu = 0$, the above integral then implies

$$0 = \int \frac{\bar{\mathrm{d}}^3 p}{2E_{\vec{p}}}p^\mu\partial_\mu f = \partial_\mu j^\mu\,, \tag{4.27}$$

where we took the partial derivative out of the integral and used Eq. (4.9). Therefore, the particle current is conserved.

Instead of setting $b_\mu = 0$, we can set $a = 0$ to arrive at

$$0 = \int \frac{\bar{\mathrm{d}}^3 p}{2E_{\vec{p}}}p^\nu p^\mu\partial_\mu f = \frac{1}{2}\partial_\mu T^{\mu\nu}\,, \tag{4.28}$$

where we took the partial derivative out of the integral and used the definition in Eq. (4.13). This establishes conservation of the tensor,

$$T^{\mu\nu} = \int \frac{\bar{\mathrm{d}}^3 p}{2E_{\vec{p}}}p^\nu p^\mu f\,, \tag{4.29}$$

which is the energy-momentum tensor of the system of particles with distribution function $f$.

## 4.4 Local equilibrium and perfect fluid

The fluid is in local equilibrium in the presence of collisions when $R(p_1,x) = \bar{R}(p_1,x)$, i.e. $C[f] = 0$. From Eqs. (4.22) and (4.23) we find that the fluid is in local equilibrium when, $\forall\, p_1, p_2, p_1', p_2'$,

$$f_1 f_2 = f_1' f_2' \tag{4.30}$$

$$\Rightarrow \quad \ln f_1 + \ln f_2 = \ln f_1' + \ln f_2'\,, \tag{4.31}$$

where we introduced the short-hand notation $f_i^{(\prime)} = f(p_i^{(\prime)}, x)$. This implies that the quantity $\ln f_1 + \ln f_2$ is conserved. Therefore it must be expressible in terms of the conserved quantities of the system. According to Eq. (4.26), it can therefore be written as

$$\ln f^{\mathrm{eq}}(p, x) = a(x) + b_\mu(x) p^\mu, \tag{4.32}$$

in local equilibrium. We rewrite $a(x)$ and $b_\mu(x)$ in a suggestive notation, $a(x) = \beta(x)\mu(x)$ and $b_\mu(x) = \beta(x) u_\mu(x)$, with $u^2 = -1$, and we recover the classical equilibrium distribution function

$$f^{\mathrm{eq}}(p, x) = \exp\left[\beta(x)(p \cdot u(x) + \mu(x))\right]. \tag{4.33}$$

We then see that $\beta(x) = 1/T$ is the inverse temperature, $\mu(x)$ the chemical potential, and $u^\mu(x)$ is the local 4-velocity of the system of particles which we can now start calling a fluid. In the fluid local rest frame, $u^\mu = (1, \vec{0})^\mu$, $p \cdot u = -p^0$, where $p^0 = E_{\vec{p}}$ is the particle energy.

To extend this analysis to quantum scattering, we need to account for the fact that two fermions cannot occupy the same quantum state (Fermi blocking), and that bosonic wave functions add coherently (Bose enhancement). This can be done by writing the particle production and destruction rates as

$$R(p_1, x) = \int \frac{\bar{\mathrm{d}}^3 p_2}{2E_2} \frac{\bar{\mathrm{d}}^3 p_1'}{2E_1'} \frac{\bar{\mathrm{d}}^3 p_2'}{2E_2'} f_1 f_2 (1 \pm f_1')(1 \pm f_2') \overrightarrow{W}, \tag{4.34}$$

$$\bar{R}(p_1, x) = \int \frac{\bar{\mathrm{d}}^3 p_2}{2E_2} \frac{\bar{\mathrm{d}}^3 p_1'}{2E_1'} \frac{\bar{\mathrm{d}}^3 p_2'}{2E_2'} f_1' f_2' (1 \pm f_1)(1 \pm f_2) \overleftarrow{W}. \tag{4.35}$$

The additional factors of $1 \pm f_i^{(\prime)}$ implement the Bose enhancement and Fermi blocking in $C[f]$.

In local equilibrium, the particle production and destruction rates are equal, i.e. $C[f] = 0$. Hence, the distribution function has to satisfy

$$f_1 f_2 (1 \pm f_1')(1 \pm f_2') = f_1' f_2' (1 \pm f_1)(1 \pm f_2). \tag{4.36}$$

Separating primed and unprimed variables reveals that $\ln f_1/(1 \pm f_1) + \ln f_2/(1 \pm f_2)$ is conserved. As for the classical case, this implies that the distribution function in equilibrium can be written as

$$\ln \frac{f^{\mathrm{eq}}}{1 \pm f^{\mathrm{eq}}} = a + b_\mu p^\mu, \tag{4.37}$$

where it is understood that as before $a$ and $b$ depend on $x$. Rearranging and rewriting $a$ and $b$ in terms of the more physical quantities $\mu$, $\beta$ and $u$ yields the following expression for the distribution function,

$$f^{\mathrm{eq}} = \frac{1}{e^{\beta(\mu + u \cdot p)} \pm 1}. \tag{4.38}$$

In the early universe the chemical potential is negligible, $\mu \simeq 0$. Then, the energy-momentum tensor in local equilibrium reads

$$T^{\mu\nu} = (e + p) u^\mu u^\nu + p g^{\mu\nu}, \tag{4.39}$$

where all quantities depend on $x$. This is the energy-momentum tensor for a perfect fluid with energy density $e$ and pressure[3] $p$. Despite an ambiguity in the notation, the pressure is not to be confused with 4-momentum.

---

[3] The pressure is defined as

$$p = \int \frac{\mathrm{d}^3 p}{(2\pi)^3} \frac{\vec{p}^2}{2p^0} f(p, x). \tag{4.40}$$

# 5 Hydrodynamics with field-dependent mass

In the early universe the value of the Higgs changes in time during the electroweak phase transition. The Higgs transitions from the symmetric to the broken phase. As discussed in Sec. 3, the particle masses depend on the value of the Higgs. Since the transition does not happen at the same time in the entire universe, the particle masses are space-time dependent due to their field dependency. In this section, we generalize the results of the previous section to the case of a fluid of particles with space-time dependent mass.

The theory of hydrodynamics with a field dependent mass is structurally similar to hydrodynamics with electromagnetic forces and hence to the wide field of plasma physics. This is because the field dependency of the mass leads to a non-zero external force $F^\mu$. In the case of plasma physics the motion of particles under Lorentz forces is described by the Vlasov equation. In close analogy, we will derive Boltzmann's equation for a fluid with field dependent (and hence space-time dependent) mass in the following.

We start from the action for a single particle (note this is not the action for the field $\phi$)

$$S = -\int d\tau\, m \sqrt{-g_{\mu\nu} \frac{dx^\mu}{d\tau} \frac{dx^\nu}{d\tau}}\,, \tag{5.1}$$

where $m = m(\phi(x(\tau)))$ is the field-dependent mass, $x^\mu = x^\mu(\tau)$ is the space-time coordinate of a particle, parameterised by the particle's proper time $\tau$, and $g_{\mu\nu}$ is the space-time metric. Varying this action we obtain the equation of motion

$$\frac{d}{d\tau}\left(m\frac{dx^\mu}{d\tau}\right) + \partial^\mu\phi\frac{dm}{d\phi} = 0\,. \tag{5.2}$$

From this equation we identify the 4-momentum $p^\mu = m\, dx^\mu/d\tau$, and the force

$$F^\mu = -\partial^\mu\phi\frac{dm}{d\phi}\,, \tag{5.3}$$

where we used the equation of motion (5.2). Hence, the field-dependence of the mass yields a force acting on the particle. This has to be taken into account when deriving conservation laws.

In the previous section, we derived the conservation of the particle current and energy-momentum assuming there are no external sources. Now, consider instead the Boltzmann equation with collisions and external forces,

$$\left(p^\mu\partial_\mu + mF^\mu\frac{\partial}{\partial p^\mu}\right)\Theta(p^0)\delta(p^2 + m^2)f = C[f]\,, \tag{5.4}$$

where we have introduced the on-shell condition again. We multiply both sides with $p^\nu$, integrate over momenta, and use Eq. (4.26) to find

$$0 = \int \frac{d^4p}{(2\pi)^4} p^\nu C[f] \tag{5.5}$$

$$= \int \frac{d^4p}{(2\pi)^4} p^\nu \left(p^\mu\partial_\mu + mF^\mu\frac{\partial}{\partial p^\mu}\right)\Theta(p^0)\delta(p^2 + m^2)f \tag{5.6}$$

$$= \partial_\mu \int \frac{d^3p}{(2\pi)^3} \frac{p^\mu p^\nu}{2E_{\vec{p}}} f - mF^\nu \int \frac{d^4p}{(2\pi)^4}\Theta(p^0)\delta(p^2 + m^2)f \tag{5.7}$$

$$= \frac{1}{2}\partial_\mu T^{\mu\nu} - mF^\nu \int \frac{d^3p}{(2\pi)^3}\frac{1}{2E_{\vec{p}}}f\bigg|_{p^0 = E_{\vec{p}}}\,, \tag{5.8}$$

where we assumed that collisions are local and preserve momenta. For the last step, we used the definition of the energy-momentum tensor as in Eq. (4.13), integrated the second term by parts and used that $\partial p^\nu/\partial p^\mu = \delta^\nu_\mu$. Using Eq. (5.3), this yields

$$\partial_\mu T^{\mu\nu} = -\partial^\nu \phi \frac{\mathrm{d}m^2}{\mathrm{d}\phi} \int \frac{\mathrm{d}^3 p}{(2\pi)^3} \frac{1}{2E_{\vec{p}}} f(p,x) \bigg|_{p^0 = E_{\vec{p}}}, \tag{5.9}$$

where the 4-momentum is constrained such that $p^0 = E_{\vec{p}}$. The energy-momentum tensor of the fluid is not conserved when the mass is field-dependent, but sourced by a term proportional to the change of the mass throughout spacetime. Total energy-momentum of the fluid-field system is conserved, so there is a corresponding term in the energy-momentum conservation equation for the field.

Note that in the derivation of Eq. (5.9) we assumed momentum conservation in collisions. However, in the presence of gradients in the scalar field it is not clear if momentum is conserved; there could be exchange of momentum with the field during the collision. However, as long as the gradients in the scalar field are small, by which we mean

$$\lambda_{\mathrm{mfp}} \frac{\partial \phi}{\phi} \lesssim \mathcal{O}(1), \tag{5.10}$$

where $\lambda_{\mathrm{mfp}}$ is the mean free path in the fluid, conservation of momentum in collisions should be a good approximation. In the opposite limit, the particles are unlikely to interact in the wall, and conservation of momentum in collisions is again recovered.

## 5.1 Complete model of scalar field fluid system

A more detailed description of a system consisting of a scalar field and a fluid is given in Ref. [45]. As a starting point, split the full energy-momentum tensor into two components, one for the fluid $T_f^{\mu\nu}$ and one for the Higgs $T_\phi^{\mu\nu}$, as

$$T_f^{\mu\nu} = (e + p_1)u^\mu u^\nu + p_1 g^{\mu\nu}, \tag{5.11}$$

$$T_\phi^{\mu\nu} = \partial^\mu \phi \partial^\nu \phi - g^{\mu\nu}\left(\frac{1}{2}(\partial\phi)^2 + V_0(\phi)\right), \tag{5.12}$$

where we label the fluid pressure as $p_1$ for now. As an example we use the symmetry breaking potential

$$V_0(\phi) = \frac{\lambda}{4}\left(\phi^2 - v^2\right)^2, \tag{5.13}$$

for the scalar field. Note that there are some field-dependent terms in the fluid pressure, which we define as

$$p_1(\phi, T) = \frac{\pi^2}{90} g_{\mathrm{eff}} T^4 - V_1(\phi, T), \tag{5.14}$$

where the fluid pressure is the total contribution from all particles, equal to the negative of the free energy densities (2.19) and (2.26)

$$p_1(\phi, T) = -\sum_B f_B(m(\phi), T) - \sum_F f_F(m(\phi), T). \tag{5.15}$$

The full thermally corrected potential then reads

$$V_T(\phi) = V_0(\phi) + V_1(\phi, T). \tag{5.16}$$

In the high-temperature expansion, $V_1(\phi, T)$ contains all $T$-dependent terms of the potential in Eq. (3.7).

We demand that the full energy-momentum tensor is conserved,

$$\partial_\mu \left( T_f^{\mu\nu} + T_\phi^{\mu\nu} \right) = 0 , \tag{5.17}$$

and hence non-conservation of $T_f^{\mu\nu}$ has to appear in $T_\phi^{\mu\nu}$. Using Eq. (5.9) this implies

$$\partial_\mu T_\phi^{\mu\nu} = +\partial^\nu \phi \frac{dm^2}{d\phi} \int \frac{d^3 p}{(2\pi)^3} \frac{1}{2E_{\vec{p}}} f(p, x) . \tag{5.18}$$

The energy-momentum tensor for the field is hence not conserved individually, fluid contributions can source changes in its energy and momentum.

Computing the equation of motion for the scalar field one obtains

$$\Box \phi - V_0'(\phi) = -\frac{dm^2}{d\phi} \int \frac{d^3 p}{(2\pi)^3} \frac{1}{2E_{\vec{p}}} f(p, x) , \tag{5.19}$$

where prime denotes the derivative with respect to the argument $\phi$. Here again the right-hand side stems from the fluid contributions and in a more complicated theory should contain a sum over all massive degrees of freedom.

Let us study the situation, when the system is close to equilibrium and parametrize the distribution function as a small perturbation around equilibrium as $f(p, x) = f^{\mathrm{eq}}(p, x) + \delta f(p, x)$. If the fluid were in local equilibrium everywhere, Eq. (5.19) becomes [45]

$$\Box \phi - V_0'(\phi) = -\frac{dm^2}{d\phi} \int \frac{d^3 p}{(2\pi)^3} \frac{1}{2E_{\vec{p}}} f^{\mathrm{eq}}(p, x) = V_1'(\phi, T) \tag{5.20}$$

according to the definition of the free energies in Eqs. (2.18) and (2.26). Therefore, the equation of motion of the Higgs in exact equilibrium reads,

$$\Box \phi - V_T'(\phi) = 0 . \tag{5.21}$$

Consequently the equation for small departures from fluid equilibrium (5.19) becomes

$$\Box \phi - V_T'(\phi) = -\frac{dm^2}{d\phi} \int \frac{d^3 p}{(2\pi)^3} \frac{1}{2E_{\vec{p}}} \delta f(p, x) \tag{5.22}$$

to first order in $\delta f$.

Let us repackage the contribution from the effective potential into the fluid by defining the fluid pressure as $p(\phi, T) = p_1(\phi, T) - V_0(\phi)$. The new fluid energy-momentum tensor then reads

$$T_f^{\mu\nu} = (e + p)u^\mu u^\nu + p g^{\mu\nu} , \tag{5.23}$$

which yields

$$\partial_\mu T_f^{\mu\nu} + V_T'(\phi)\partial^\nu \phi = -\partial^\nu \phi \frac{dm^2}{d\phi} \int \frac{d^3 p}{(2\pi)^3} \frac{1}{2E_{\vec{p}}} \delta f(p, x) \tag{5.24}$$

for its conservation equation. The right hand side is generated by departures from the equilibrium phase-space distribution $f^{\mathrm{eq}}$, which in turn have to be induced by gradients of the scalar field. We assume that these are small, in the sense outlined at the end of the last section.

To express the dependence on the gradient in analogy to linear response theory, we write the integral of the perturbed distribution function on the right hand side of Eq. (5.24) as a term proportional to the gradient of the scalar field,

$$\int \frac{d^3 p}{(2\pi)^3} \frac{1}{2E_{\vec{p}}} \delta f(p, x) = \tilde{\eta} u^\mu \partial_\mu \phi , \tag{5.25}$$

where we use the fluid 4-velocity $u^\mu$ to contract the open index in order to respect isotropy. The proportionality factor $\tilde\eta$ might in general depend on the field value, the temperature, and the fluid $\gamma$-factor, or $\tilde\eta = \tilde\eta(T, \phi, \gamma)$. The overall mass dimension of $\tilde\eta$ is zero, so it must depend on the ratio $\phi/T$.

This leaves us with the equation of motion

$$\Box\phi - V_T'(\phi) = -\tilde\eta \frac{\mathrm{d}m^2}{\mathrm{d}\phi} u^\mu \partial_\mu \phi \,. \tag{5.26}$$

Note that in using the ansatz (5.25) we assumed full Lorentz symmetry. However, invariance under boosts is broken in a plasma. The following argument from entropic considerations supports the Lorentz-symmetric form.

Consider the scalar product of $u$ with both sides of Eq. (5.24),

$$u_\nu \partial_\mu (w u^\mu u^\nu + p g^{\mu\nu}) + V_T'(\phi) u \cdot \partial \phi = \tilde\eta (u \cdot \partial \phi)^2, \tag{5.27}$$

where $w = e + p = Ts$ is the enthalpy density, and $s = \mathrm{d}p/\mathrm{d}T$ is the entropy density. From this we can derive a conservation equation for the entropy current $S^\mu = su^\mu$,

$$\partial \cdot S = \frac{\tilde\eta}{T}(u \cdot \partial \phi)^2. \tag{5.28}$$

The fact that this is always positive supports the claim that the term on the right hand side of Eq. (5.25) is indeed proportional to $u \cdot \partial \phi$. In this model entropy is generated by the interaction of the field and the fluid.

# 6 The bubble nucleation rate

We now to study how, and how quickly, a phase transition happens. We focus on first-order phase transitions, which are characterised by a critical temperature, a latent heat, and mixed phases separated by phase boundaries with a characteristic surface energy. An *order parameter* distinguishes between the phases.

A simple example is the phase transition between water in its liquid and vapour form, at a critical temperature of 373 K at 1 atmosphere pressure. The order parameter is the density, which changes by a factor of about 1000 at the phase transition.

In the laboratory, systems cooled below their critical temperature, proceed by the nucleation of small bubbles or droplets of the new phase around impurities. Especially pure systems can be cooled well below their critical temperature before the transition occurs.

The early Universe is thought to be perfectly pure, so bubbles of the new phase can appear only by thermal or quantum fluctuations.[4] The order parameter is a scalar field $\phi$, and we will take the values of the field in the two phases to be $\phi = 0$ in the high temperature phase and $\phi = \phi_\mathrm{b}$ in the low temperature phase.

The transition proceeds via bubble nucleation: individual bubbles of the new ground state $\phi = \phi_\mathrm{b}$ appear due to thermal fluctuations. If the temperature of the universe is below the critical value $T_\mathrm{c}$, these bubbles grow if they exceed a critical radius $R_\mathrm{c}$. Eventually they merge, and the whole universe is in the new phase. In this section we will compute the rate at which these bubbles appear and how the system changes from the old to the new ground state. This process was first studied in the context of statistical mechanics in Ref. [48], and in quantum field theory in Refs. [49] and [50]. In turns out that at high temperatures higher than the

---

[4]This picture may not be accurate if there are primordial black holes, which can act as nucleation sites [46,47].

masses of the particles involved, the quantum field theory is well approximated by statistical mechanical description.

We start with the partition function of a classical field $\phi$ at temperature $\beta = 1/T$, which is the functional integral

$$Z_\beta = \int \mathcal{D}\pi \mathcal{D}\phi\, e^{-\beta H}, \tag{6.1}$$

with the Hamiltonian

$$H = \int \mathrm{d}^3 x \left[ \frac{1}{2}\pi^2 + \frac{1}{2}(\vec{\nabla}\phi)^2 + V_T(\phi) \right], \tag{6.2}$$

where $\pi$ is the conjugate momentum to the field $\phi$. We will use the partition function to calculate the rate per unit volume of fluctuations over a thermal barrier in the potential $V_T$.

## 6.1 Transition rate in 0 dimensions

To get started, we discuss a simple toy model without spatial dimensions [51]. Therefore, spatial gradients $\vec{\nabla}\phi$ do not exist, and the integrals over functions $\pi$ and $\phi$ become ordinary integrals. The theory describes a particle with coordinate $\phi$ in a potential $V_T$ as depicted in Fig. 8. The partition function for this simple system is

$$Z_\beta = \int \mathrm{d}\pi\, \mathrm{d}\phi\, e^{-\beta\left[\pi^2/2 + V_T(\phi)\right]}. \tag{6.3}$$

We can immediately perform the integral over $\pi$ yielding

$$Z_\beta = \sqrt{\frac{2\pi}{\beta}} \int \mathrm{d}\phi\, e^{-\beta V_T(\phi)}. \tag{6.4}$$

To carry out the remaining integration over $\phi$ we use the saddle point approximation. In principle the potential $V_T(\phi)$ has three stationary points, two minima, a local one at $\phi = 0$ and a global one at $\phi = \phi_\mathrm{b}$ as well as a maximum at $\phi = \phi_\mathrm{m}$ separating the two minima (see Fig. 8). However, in our set-up we assume that initially there is a thermal bath of particles only in the metastable ground state at $\phi = 0$. To infer the transition rate of particles fluctuating *to* the stable ground state at $\phi = \phi_\mathrm{b}$, only two saddle-points are of relevance: the false minimum and the maximum. Intuitively, this can be understood from the fact that once a particle in the metastable ground state makes it over the top of the potential barrier, due to thermal fluctuations, it just rolls down into the true ground state. Hence, the transition rate is dominated by the two aforementioned saddle-points. In particular, even though with time particles settle in the global minimum, the flux of particles from the stable to the metastable ground state will always be negligible.

Let us evaluate the partition function at the two saddle points. The first point sits at $\phi = 0$. We hence expand as $\phi = 0 + \delta\phi$ to obtain

$$\begin{aligned} Z_\beta^0 &= \sqrt{\frac{2\pi}{\beta}} \int \mathrm{d}\phi \exp\left[ -\beta\left( V_T(0) + \frac{1}{2}V_T''(0)\delta\phi^2 \right) \right] \\ &= \sqrt{\frac{2\pi}{\beta}}\sqrt{\frac{2\pi}{\beta V_T''(0)}} e^{-\beta V_T(0)}. \end{aligned} \tag{6.5}$$

The second stationary point is at $\phi = \phi_\mathrm{m}$ with $V_T''(\phi_\mathrm{m}) < 0$. As the second derivative at the stationary point is negative here we need to deform the integration contour into the complex

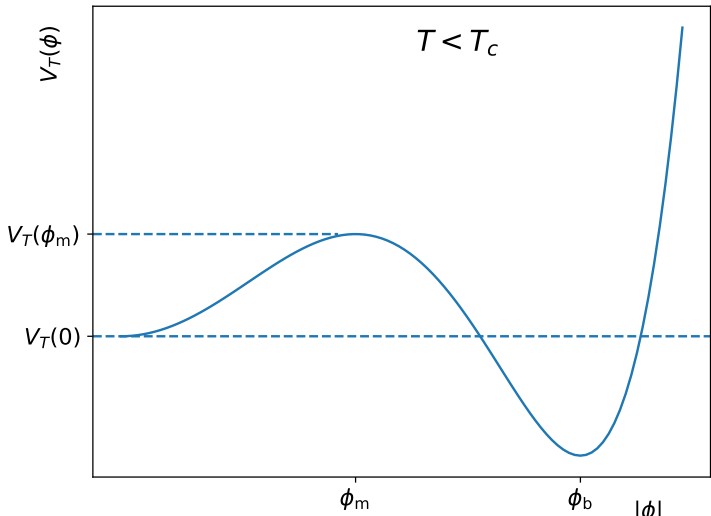

Figure 8: The effective thermal potential $V_T(\phi)$ for a temperature below the critical temperature, $T < T_c$. The potential has two minima $\phi = 0$ an $\phi = \phi_b$, and a maximum at $\phi = \phi_m$, which we use for the saddle point approximation. Due to thermal fluctuations, the field can transition from the symmetric phase to the Higgs phase.

plane, which results in the integral acquiring an imaginary part. At this second stationary point we obtain, for the imaginary part,

$$Z_\beta^1 = \sqrt{\frac{2\pi}{\beta}} \sqrt{\frac{2\pi}{V_T''(\phi_m)}} \frac{1}{2} e^{-\beta V_T(\phi_m)}. \tag{6.6}$$

This expression is imaginary because of the negative sign for $V_T''$. The factor of $1/2$ is due to the integration contour transversing only one half of the complex plane. The leading term in the full partition function is the sum of these two terms, $Z_\beta = Z_\beta^0 + Z_\beta^1$.

Computing the free energy we obtain

$$F = -\frac{1}{\beta} \ln Z \approx F_\beta^0 \pm \frac{i}{2\beta} e^{-\beta \Delta V_T} \sqrt{\frac{V_T''(0)}{\left|V_T''(\phi_m)\right|}}, \tag{6.7}$$

where $\Delta V_T \equiv V_T(\phi_m) - V_T(0)$ is the barrier height in the potential, and we expanded the logarithm for $\beta \Delta V_T \gg 1$. The two signs arise due to the choice of the integration contour.

Next, we compute the the probability flux $\Gamma$ across the potential barrier given by

$$\Gamma = \frac{1}{Z_\beta} \int d\pi \, d\phi \, e^{-\beta H} \delta(\phi - \phi_m) \pi \Theta(\pi). \tag{6.8}$$

The $\delta$-distribution picks the saddle point at the maximum for us. The Heaviside function only allows for positive velocities (increasing field values) and hence we only consider particles that flow to the new minimum. The whole expression describes a flux of phase space volume from the old to the new ground state. Evaluating Eq. (6.8) in a similar manner as before yields

$$\Gamma = \frac{1}{2\pi} \sqrt{V_T''(0)} e^{-\beta \Delta V_T}. \tag{6.9}$$

Comparing this equation to the expression for the free energy (6.7) establishes the relation

$$\Gamma = \frac{\beta}{\pi} \sqrt{V_T''(\phi_{\mathrm{m}})} \, |\mathrm{Im}\{F\}| \,. \tag{6.10}$$

The probability flux density is thus proportional to the imaginary part of the free energy. To be explicit, it is given by

$$\mathrm{Im}\{F\} = \frac{T}{2} \sqrt{\frac{V_T''(0)}{\left|V_T''(\phi_{\mathrm{m}})\right|}} \, e^{-\beta \Delta V_T} \,. \tag{6.11}$$

Note that in Eq. (6.8) we implicitly assumed that the particles were moving freely. This is equivalent to assuming that the particles do not interact with the heat bath which maintains thermal equilibrium as they cross the barrier. The timescale for crossing the barrier can be estimated as $\tau_{\mathrm{m}} = 1/\sqrt{\left|V_T''(\phi_{\mathrm{m}})\right|}$. If the mean free time between interactions is much less than $\tau_{\mathrm{m}}$, then it is more appropriate to study the diffusion of particles across the barrier, which can be modelled by the Langevin equation

$$\dot{\pi} = -\gamma \pi - V_T'(\phi) + \xi(t). \tag{6.12}$$

Here, $\gamma$ is the diffusion constant, and $\xi(t)$ represents the forces exerted by the heat bath. It is a Gaussian random variable, obeying $\langle \xi(t)\xi(t')\rangle = 2\gamma T \delta(t-t')$, meaning that it is uncorrelated from one instant to the next, but has a mean amplitude $\sqrt{2\gamma T}$. In the case $\gamma \tau_{\mathrm{m}} \gg 1$, the rate of particles crossing over the barrier is

$$\Gamma = \frac{1}{2\pi} \frac{\sqrt{V_T''(0)\left|V_T''(\phi_{\mathrm{m}})\right|}}{\gamma} e^{-\beta \Delta V_T} \,. \tag{6.13}$$

This is a factor $1/\gamma \tau_{\mathrm{m}}$ smaller than the rate of free particles crossing the barrier.

The problem of how classical particles escape over an energy barrier was originally solved by Kramers [52] in the context of chemical reactions. The formulation of the Kramers escape problem in quantum field theory was recently studied in Ref. [53]. For an early reference relating the imaginary part of the free energy to the tunneling rate, see Ref. [54].

## 6.2 Bubble nucleation in 3 dimensions

After having derived the probability flux without spatial dimensions, in this section we will repeat this procedure for 3 dimensions.

### 6.2.1 The critical bubble

The starting point is the path integral which for a thermal field theory is given by

$$\mathcal{Z}_\beta = \int \mathcal{D}\pi \int \mathcal{D}\phi \, e^{-\beta H[\pi,\phi]} \,, \tag{6.14}$$

where the Hamiltonian $H$ is given by Eq. (6.2) now including spatial gradients of the scalar field. We can perform the integration over $\pi$, as it is Gaussian, leading to

$$\mathcal{Z}_\beta = \mathcal{N} \int \mathcal{D}\phi \, e^{-\beta E_T[\phi]} \,, \tag{6.15}$$

where $\mathcal{N}$ is the result of the integration. It will play no role in the following.

We will again perform a saddle point evaluation of this integral, which means first identifying the stationary point of the function in the exponential.

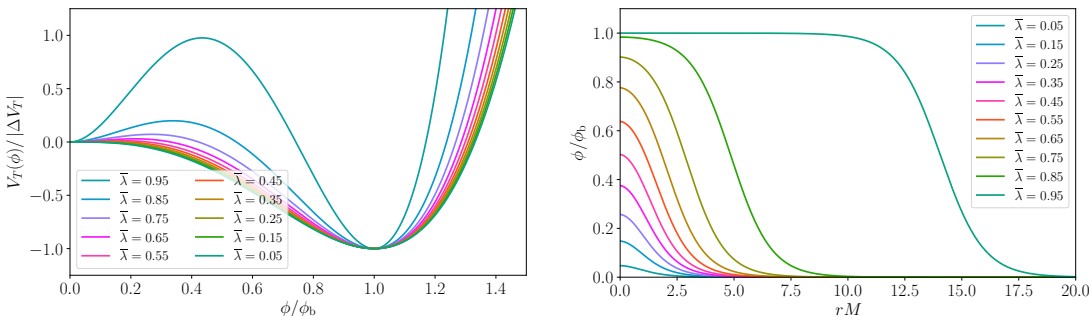

Figure 9: *Left panel:* the scaled quartic potential as a function of the parameter $\bar{\lambda}$, defined in Eq. (6.18). *Right panel:* the critical bubble solution to Eq. (6.17). Scripts by kind permission of D. Cutting, see also Ref. [55].

At the stationary or saddle point, we have $\delta E_T[\phi]/\delta\phi = 0$, leading to the following differential equation

$$-\vec{\nabla}^2\phi + \frac{\partial V_T}{\partial\phi} = 0\,. \tag{6.16}$$

It turns out that the energy functional $E_T$ is minimized for a radially symmetric field configuration. In that case, the above equation of motion can be rewritten as

$$-\frac{1}{r^2}\frac{\mathrm{d}}{\mathrm{d}r}\left(r^2\frac{\mathrm{d}\phi}{\mathrm{d}r}\right) + V'_T(\phi) = 0\,. \tag{6.17}$$

We will suppose a thermal effective potential of the form sketched in Fig. 8, and assume a high temperature approximation of quartic form as in Eq. (3.8). We are looking for a solution which represents a finite-energy fluctuation away from the metastable phase $\phi = 0$ at a temperature just below $T_c$. The asymptotics of the solution are therefore: as $r \to \infty$, $\phi \to 0$, and as $r \to 0$, $\phi'(r) \to 0$, where the prime denotes derivative with respect to $r$.

A trivial solution $\phi(r) = 0$ always exists. A non-trivial solutions can be found by numerical integration, using a shooting method (see e.g. Ref. [55]). Although there are three parameters, it turns out that the differential equation can be rewritten so that it depends only on the combination

$$\bar{\lambda} = \frac{9\lambda D}{2A^2}\,, \tag{6.18}$$

provided one scales the potential with the vacuum energy difference, the field with its stable phase value $\phi_b$, and radial distance with the effective mass $M = D\sqrt{T^2 - T_0^2}$. The results for selected values of $\bar{\lambda}$ are shown in Fig. 9. We refer to a non-trivial solution of Eq. (6.17) with these boundary conditions as a *critical bubble* and denote it by $\bar{\phi}(r)$.

At the critical temperature, $\bar{\lambda} \to 1$, and the potential is approximately

$$V(\phi) = \frac{\lambda}{4}\phi^2(\phi - \phi_b)^2\,. \tag{6.19}$$

This leads to the following approximate solution to the equation of motion,

$$\bar{\phi} = \frac{\phi_b}{2}\left(1 - \tanh\left(\frac{r - R_c}{\ell_w}\right)\right)\,, \tag{6.20}$$

where $R_c$ is the radius of the critical bubble, $\ell_w = 1/M(T_c)$ the thickness of the bubble wall, and $R_c \gg \ell_w$. This solution is plotted in the right panel of Fig. 9 for different values of $\bar{\lambda}$. We refer to this approximate solution as a *thin-wall* bubble.

The energy of a critical bubble can be computed from the energy functional

$$E_T[\phi] = \int d^3x \left( \frac{1}{2} \left( \vec{\nabla} \phi \right)^2 + V_T(\phi) \right). \tag{6.21}$$

This is potentially divergent in an infinite volume system. We are actually interested in the difference between the energy of the field configuration of the critical bubble $\bar{\phi}$ and the energy of the metastable state $\phi = 0$, or

$$E_c = E_T[\bar{\phi}] - E_T[0]. \tag{6.22}$$

This quantity is finite, and it is the energy a thermal fluctuation needs to have in order to lift the field over the barrier in the spherical region of radius $R$.

It is simplest to analyse this expression for a thin-wall. In that case, the interior of the bubble represents the 'new' minimum (for instance the Higgs phase in the case of electroweak symmetry breaking), the exterior the 'old' minimum (i.e. the metastable phase) and the wall of the bubble defines the region where the field interpolates between the two minima.

If $\epsilon = \Delta V_T / V_T(\phi_m) \ll 1$, where $\Delta V_T = V_T(0) - V_T(\phi_b)$ is the free-energy difference between the two phases, there exists an approximate solution for $E_c$ as a function of radius $R$, given by

$$E_c(R) = 4\pi\sigma R^2 - \frac{4}{3}\pi\Delta V_T R^3. \tag{6.23}$$

Here, $\sigma$ denotes the surface tension given by

$$\sigma = \frac{2^{3/2}}{3^4} \frac{A^3}{\lambda^{5/2}} T_c^3, \tag{6.24}$$

with $A$ and $\lambda$ the constants appearing in the thermal Higgs potential (3.8). It is possible to introduce a critical radius $R_c$ at which the bubble does neither shrink nor grow. This is the radius where the force, due to the release of latent heat, pushing the bubble to grow, is balanced with the interaction of the bubble with the plasma of particles creating friction. Above the critical radius it is favorable for the bubble to grow. The critical radius is defined as the stationary point of the energy $E$ as a function of the radius $R$, i.e.

$$\frac{\partial E}{\partial R} = 8\pi\sigma R - 4\pi\Delta V_T R^2 \bigg|_{R=R_c} \overset{!}{=} 0, \tag{6.25}$$

such that the critical radius is given by

$$R_c = \frac{2\sigma}{\Delta V_T} = \frac{2\sigma}{V_T(\phi_m)\epsilon}. \tag{6.26}$$

Plugging this expression into Eq. (6.23), we find the energy of the critical bubble as

$$E_c = E(R_c) = \frac{16\pi}{3} \frac{\sigma^3}{\Delta V_T^2}. \tag{6.27}$$

In the thin-wall approximation, the energy of the critical bubble can be computed as a function of temperature $T$ as [56]

$$E_c(T) = \frac{16\pi}{3} \frac{\sigma T_c^2}{[L(T_c - T)]^2}, \tag{6.28}$$

where $L$ is the latent heat. It is given by $L = T_c(p_s - p_b)$ with $p_s$ and $p_b$ the fluid pressure in the symmetric and in the broken phase, resp. Hence, the energy $E_c$ diverges at the critical temperature.

Note the different uses of the word "critical" in critical bubble, with radius $R_c$ and energy $E_c$, and the critical temperature $T_c$.

### 6.2.2 Saddle point evaluation

We now return to completing the saddle-point approximation of the integral (6.15). We recall that the saddle point method consists in expanding around stationary values of the energy functional $E_T[\phi]$. In the case at hand the two previously mentioned classical solutions to the equations of motion, the trivial solution $\phi(r) = 0$ and the critical bubble solution $\phi(r) = \bar{\phi}(r)$, extremise $E_T[\phi]$. In contrast to the simplified case without spatial dimensions where we evaluated the partition function at the extrema of the thermal potential, i.e., at the *points* $\phi = 0$ and $\phi = \phi_m$, we now evaluate the partition function at two classical *solutions* out of the field configuration space of $\phi$. So the partition function will be a sum of two terms $\mathcal{Z}^0$ and $\mathcal{Z}^1$ evaluated around the extrema $\phi = 0$ and the critical bubble $\phi = \bar{\phi}$, respectively.

Let us first expand around the classical solution $\bar{\phi}$ as $\phi = \bar{\phi} + \delta\phi$. Performing the actual saddle-point approximation we find

$$\mathcal{Z} \simeq \mathcal{N}\mathcal{D}(\delta\phi)e^{-\beta E[\bar{\phi}] - \beta \int \delta\phi(-\vec{\nabla}^2 + V_T''(\bar{\phi}))\delta\phi/2}. \tag{6.29}$$

To evaluate the functional integral it is useful to diagonalise the operator $-\vec{\nabla}^2 + V_T''(\bar{\phi})$. In order to do so we consider the following eigenvalue equation

$$(-\vec{\nabla}^2 + V_T''(\bar{\phi}))\psi_\alpha = \lambda_\alpha \psi_\alpha, \tag{6.30}$$

with $\delta\phi = \sum_\alpha c_\alpha \psi_\alpha$, where the $\psi_\alpha$ are normalized so that $\beta \int d^3x |\psi_\alpha|^2 = 1$. The measure then takes the following form

$$\mathcal{D}(\delta\phi) = \int \prod_\alpha \frac{dc_\alpha}{\sqrt{2\pi}}. \tag{6.31}$$

For the non-zero eigenvalues, we have to perform Gaussian integrals, resulting in

$$\mathcal{Z}^1 \simeq \mathcal{N}' \int \prod_i \frac{dc_i}{\sqrt{2\pi}} \prod_\alpha' \frac{1}{\lambda_\alpha^{\frac{1}{2}}} e^{-\beta E[\bar{\phi}]}, \tag{6.32}$$

where the index $i$ labels the zero eigenvalues, and the prime indicates that the zero modes have been omitted from the product.

There are in fact three eigenfunctions with zero eigenvalues. Let us recall the equation of motion for a solution $\bar{\phi}$, given by

$$-\vec{\nabla}^2\bar{\phi} + V_T'(\bar{\phi}) = 0. \tag{6.33}$$

Taking the spatial gradient yields

$$\left(-\vec{\nabla}^2 + V_T''(\bar{\phi})\right)\partial_i\bar{\phi} = 0. \tag{6.34}$$

These are the *translational zero modes*, as they correspond to the change in the field when performing a small translation $x^i \to x^i + \delta x^i$, for which

$$\delta\bar{\phi} = \delta x^i \partial_i\bar{\phi}. \tag{6.35}$$

For these modes, the normalised integration variables are

$$dc^i = dx^i \left[\frac{\beta}{3}\int d^3x \left(\vec{\nabla}\bar{\phi}\right)^2\right]^{\frac{1}{2}}, \tag{6.36}$$

where we have used the spherical symmetry of the solution. One can show from the stationarity of the solution under a scale transformation $\delta\bar{\phi} = x^i\partial_i\bar{\phi}$ that the integral in the square brackets in Eq. (6.36) is equal to the energy of the critical bubble. Hence

$$dc^i = dx^i(\beta E_c)^{\frac{1}{2}}, \tag{6.37}$$

from which it follows that

$$\mathcal{Z}^1 \simeq \mathcal{N}'\mathcal{V}\left(\frac{\beta E_{\mathrm{c}}}{2\pi}\right)^{3/2}\prod_\alpha{}'\frac{1}{\lambda_\alpha^{\frac{1}{2}}}e^{-\beta E[\bar\phi]}, \tag{6.38}$$

where $\mathcal{V}$ is the volume of the system.

Furthermore, also the trivial solution $\phi = 0$ is an extremum. Therefore, if we solve Eq. (6.15) at $\phi = 0$ we obtain

$$\mathcal{Z}^0 \simeq \mathcal{N}'\prod_\alpha\frac{1}{\left(\lambda_\alpha^0\right)^{1/2}}e^{-\beta E[0]}. \tag{6.39}$$

The solution $\phi = 0$ has no zero modes, as it is translation invariant.

The partition function evaluated through a saddle-point approximation is now given as a sum of only two terms, $\mathcal{Z} \simeq \mathcal{Z}^0 + \mathcal{Z}^1$.

In order to compute the transition rate from the trivial solution to the critical bubble we have to consider the imaginary part of the free energy. This comes from the expansion around the critical bubble, which has one negative eigenvalue $\lambda_-$, linked to an unstable mode (i.e. expansion or contraction of the bubble).

We can already see that the ratio $\mathcal{Z}^1/\mathcal{Z}^0$ contains an exponentially small factor $\exp(-\beta E_{\mathrm{c}})$. Therefore, we can expand the logarithm in the free energy and approximate its imaginary part as

$$\mathrm{Im}\{F\} \approx -T\frac{|\mathcal{Z}^1|}{\mathcal{Z}^0}. \tag{6.40}$$

If the interaction rate of the field with the thermal bath is small, we use the example of the particle in a potential well to infer that the probability flux per volume $\mathcal{V}$ is then given by

$$\frac{\Gamma}{\mathcal{V}} = \frac{\sqrt{V_T''(0)}}{\pi}\left[\frac{\det\left(-\vec\nabla^2 + V_T''(0)\right)}{|\det'(-\vec\nabla^2 + V_T''(\bar\phi))|}\right]^{1/2}\left(\frac{\beta E_{\mathrm{c}}}{2\pi}\right)^{3/2}e^{-\beta E_{\mathrm{c}}}, \tag{6.41}$$

where we have introduced a standard notation for the eigenvalue products. The symbol $\det'$ means that the three zero modes have been omitted, which means that the overall mass dimension of the square root of the ratio of determinants is that of the square root of the product of three eigenvalues, or 3. Hence the mass dimension of the right hand side is 4, as is appropriate for a rate per volume.

We can estimate the order of magnitude of the dimension-4 prefactor as $\left[\sqrt{V_T''(0)}\right]^4$ in a Standard-Model-like plasma. Recalling the thermal potential in the high-temperature approximation (3.8), and using Eq. (3.13), we find that $\sqrt{V_T''(0)} \sim g^2 T$ at $T_{\mathrm{c}}$, where we have used the fact that the most important couplings in the Standard Model are all of the same order of magnitude, or $g^2 \simeq \lambda \simeq y_t$. This seems to indicate that the prefactor should go as $g^8 T^4$. It turns out that for the Higgs in a Standard Model plasma, the interactions with the $W$ and $Z$ fields are rather rapid, and it evolves diffusively [57,58], as in the Kramers problem. The diffusion constant in a plasma of $W$ and $Z$ particles is $\gamma^{-1} \sim g^4\ln(1/g)$ [59], bringing an extra factor of order $g^2\ln(1/g)$.

# 7 Dynamics of expanding bubbles

In this section we will consider what happens after the bubble has nucleated and started to expand. As the bubble gets larger, it is appropriate to move to the hydrodynamic description of the system.

We would like to know how the fluid reacts to the expanding bubble, and in particular the amplitude of the shear stresses which are generated, as they are the source of the gravitational waves.

A particular quantity of interest is the speed at which the phase boundary is expanding, the so-called wall speed $v_{\mathrm{w}}$. Having good knowledge of the wall speed is important, as the flow set-up around the expanding bubble depends strongly on it, and therefore the gravitational wave power. The flow also depends on the strength of the transition parametrised by $\alpha_{\mathrm{n}}$, introduced in Section 7.3. In this section we will see how the flow around an expanding bubble depends only on $v_{\mathrm{w}}$ and $\alpha_{\mathrm{n}}$.

The fluid flow is also of importance for baryogenesis scenarios, which can be explained through a first-order electroweak phase transition [60]. A relativistic wall speed increases the chances of detecting gravitational imprints, but decreases the efficiency with which baryon number is generated [61].

## 7.1 Wall speed

For a bubble to expand, the interior pressure must be larger than the exterior one. The pressure is given by

$$p = \frac{\pi^2}{90} g_{\mathrm{eff}} T^4 - V_T(\phi), \tag{7.1}$$

where $g_{\mathrm{eff}}$ are the effective relativistic degrees of freedom. The bubble expands if $V_T(\phi_{\mathrm{b}}) < V_T(0)$, as in this case the internal pressure exceeds the external one. This can only happen below the critical temperature.

The bubble wall (which marks the boundary between the phases) must therefore move outward through the plasma. The dynamics of the field $\phi$ driving the phase transition interacting with the plasma of particles is given by

$$\Box \phi - V_T'(\phi) = -\eta_T(\phi) u^\mu \partial_\mu \phi, \tag{7.2}$$

where $\eta_T = \tilde{\eta} \mathrm{d}m^2/\mathrm{d}\phi$, cf. Eq. (5.26). This is just the standard Klein Gordon equation equipped with a friction term due to the interaction of the bubble with the plasma.

Let us now assume that the wall-speed is constant in the rest frame of the universe. Furthermore, we assume that the bubbles are macroscopic objects such that the bubble wall can be approximated to be planar. Let the wall move into the $z$-direction, and let us assume that it sets up a flow which settles down to a steady state. In the frame of the wall, the fluid moves with speed $u^\mu(z)$, and the field profile is $\phi = \phi(z)$. Let us also assume that the fluid speed changes little as it moves across the wall, so that

$$u^\mu \simeq \gamma_{\mathrm{w}}(1, 0, 0, v_{\mathrm{w}}). \tag{7.3}$$

We can then rewrite Eq. (7.2) as

$$-\partial_z^2 \phi - V_T'(\phi) = -\eta_T(\phi) \gamma_{\mathrm{w}} v_{\mathrm{w}} \partial_z \phi. \tag{7.4}$$

Multiplying both sides of the above equation with $\partial_z \phi$ and integrating over $z$, we obtain

$$\int \mathrm{d}z \left[ \frac{1}{2} \partial_z (\partial_z \phi)^2 - \frac{\mathrm{d}V_T}{\mathrm{d}z} \right] = -\int \mathrm{d}z\, \eta_T(\phi) \gamma_{\mathrm{w}} v_{\mathrm{w}} (\partial_z \phi)^2. \tag{7.5}$$

After performing the integral on the left hand side of the equation we find

$$\Delta V_T \simeq \gamma_{\mathrm{w}} v_{\mathrm{w}} \int \mathrm{d}z\, \eta_T(\phi, \gamma_{\mathrm{w}})(\partial_z \phi)^2, \tag{7.6}$$

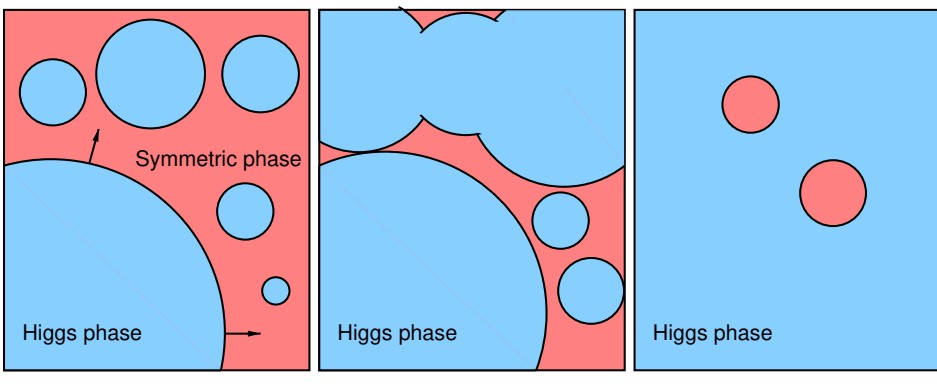

Figure 10: Due to thermal fluctuations, bubbles where the Higgs is in the stable ground state occur that expand into regions where the Higgs is still in the metastable ground state. At early times, the bubbles do not overlap, but later they combine until the entire fluid is in the stable ground state.

where $\Delta V_T = V_T(z = +\infty) - V_T(z = -\infty)$ is the pressure difference across the wall. To obtain the wall speed, one needs to solve this equation for $v_w$. In practice, it is difficult to obtain a solution. In some cases a constant-$v_w$ solution may not even exist, in which case the bubble wall must continue to accelerate. The case where $v_w \to 1$ is called *runaway* solution; the wall speed becomes ultra-relativistic [62, 63].

However, provided the product $\gamma_w \eta_T$ does not decrease with $\gamma_w$, a constant-$v_w$ solution always exists. This seems to be the generic expectation for a phase transition in a gauge theory like the Standard Model [63, 64].

In the following discussion, we assume that the transition completes in much less than a Hubble time, so that we can neglect the expansion of the universe.

In terms of $v_w$ the radius of a bubble at time $t$ that nucleated at time $t'$ is given by $R = v_w(t - t')$. Consequently, the volume $\mathcal{V}$ of this bubble is

$$\mathcal{V}(t, t') = \frac{4\pi}{3} v_w^3 (t - t')^3 . \tag{7.7}$$

In particular, this allows us to determine the fractional volume of the universe in the broken phase occupied by all bubbles. Ignoring overlaps and collisions between bubbles, the fractional volume is given by the volume of a bubble nucleated at time $t'$ multiplied by the number density of nucleated bubbles that nucleated during $(t', t' + dt')$, see Fig. 10. The latter is given by $dn(t') = \Gamma(t')dt'/\mathcal{V}$, such that the fractional volume in the metastable phase is

$$h(t) = 1 - \int_{t_c}^{t} \frac{4\pi}{3} v_w^3 (t - t')^3 \frac{\Gamma(t')}{\mathcal{V}} dt' \tag{7.8}$$

when we do not take into account overlaps. When we include overlaps, the fractional volume in the metastable phase is given by [56, 65, 66]

$$h(t) = \exp\left[ - \int_{t_c}^{t} dt' \frac{4\pi}{3} v_w^3 (t - t')^3 \frac{\Gamma(t')}{\mathcal{V}} \right] . \tag{7.9}$$

At early times, when the exponent is small, we recover the linear law as in Eq. (7.8). At late times, $h(t)$ tends rapidly to zero as expected.

Next, we consider the behaviour of the bubble nucleation rate per unit volume $\Gamma(t')/\mathcal{V}$, which governs the progress of the phase transition. It is most sensitive to the dimensionless

combination $S = E_c/T$, which decreases rapidly from infinity for $T < T_c$. Hence $\Gamma(t')/\mathcal{V}$ grows very rapidly from zero.

As a first estimate, let $t_H$ be the time where one bubble is nucleated per Hubble volume per Hubble time. This means that at $t_H$ the nucleation rate per volume $\mathcal{V}$ is given by

$$\frac{\Gamma(t_H)}{\mathcal{V}} = H^4, \tag{7.10}$$

where $H$ is the Hubble rate. The time $t_H$ can be thought of as the moment when the phase transition starts.

As a rough estimate, suppose there is a phase transition with $T_c = 100\,\text{GeV}$. We estimated previously that $\Gamma(t')/\mathcal{V} \sim T_c^4 \exp(-E_c/T_c)$, where we have dropped all dimensionless constants. The Friedmann equation tells us that the Hubble rate is $H(T_c)^2 \sim GT^4$. Hence the first bubble nucleates when $S$ has dropped to

$$S_H \sim \log(G^2 T_c^4) \sim 100. \tag{7.11}$$

The logarithm means that the value of $S$ needed for the first bubble to nucleate in a Hubble volume is insensitive to all the constants we have dropped.

As the universe continues to cool, the first bubbles grow, and new bubbles appear, which convert the old phase into the new one. Let us consider a later reference time, which we denote by $t_f$, such that $h(t_f) = 1/e$ is satisfied. That means that when $t = t_f$ roughly 64% of the universe is converted to the Higgs phase.

We now define the transition rate parameter $\beta$ as

$$\beta \equiv \frac{\mathrm{d}}{\mathrm{d}t} \log\left(\frac{\Gamma(t)}{\mathcal{V}}\right)\bigg|_{t=t_f}, \tag{7.12}$$

where the derivative is to be evaluated at $t_f$. Around this time, we can now write Eq. (6.41) after a Taylor expansion of $\log \Gamma$ as

$$\frac{\Gamma}{\mathcal{V}} = \frac{\Gamma_f}{\mathcal{V}} e^{\beta(t-t_f)}, \tag{7.13}$$

with $\Gamma_f \equiv \Gamma(t_f)$. We are now able to perform a saddle-point approximation to the integral Eq. (7.9), as

$$-\log h(t) \simeq \int^t \mathrm{d}t' \frac{4\pi}{3} v_w^3 (t-t')^3 \frac{\Gamma_f}{\mathcal{V}} e^{\beta(t'-t_f)} \tag{7.14}$$

$$= \frac{4\pi}{3} v_w^3 \frac{\Gamma_f}{\mathcal{V}} \frac{3!}{\beta^4} e^{\beta(t-t_f)}. \tag{7.15}$$

Using the definition of $t_f$, we find that

$$8\pi \frac{v_w^3}{\beta^4} \frac{\Gamma_f}{\mathcal{V}} = 1, \tag{7.16}$$

which can be used to compute $t_f$, once the form of $\Gamma(t)$ is known. Thus we find that $h(t)$ takes a very simple form, namely

$$h(t) = \exp\left[-e^{\beta(t-t_f)}\right]. \tag{7.17}$$

In Fig. (11) we show the fractional volume that is in the metastable ground state, i.e. $h(t)$, as a function of time $t$. We also plot the derivative of that quantity, which quantifies the rate at which the physical volume of the metastable ground state is changing. It turns out that

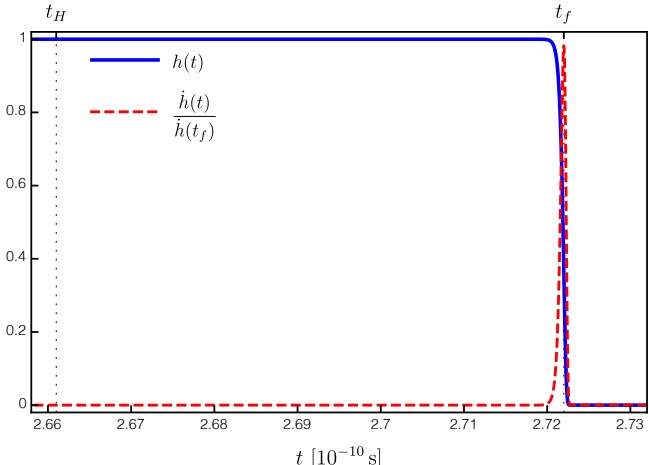

Figure 11: The time-evolution of the fractional volume in the metastable ground state, $h$, (solid blue) and its derivative $\dot{h}$ normalized to the maximum value (dashed red) is shown. At the time $t_f$ the change of the fractional volume is at its maximum, i.e. the phase transition is very fast in this moment. Shortly after, essentially the entire universe is in the new ground state. The time $t_H$ is very close before $t_f$. Their relation depends on the values of $v_w$, $\beta$ and $H$. To draw the plot, we use the realistic values of $T_H = 105.2\,\text{GeV}$, $\beta = 8950H$ and $v_w = 0.9$. Then the phase transition has completed at a temperature $T_f = 104.9\,\text{GeV}$. This choice implies that $t_f - t_H \sim 10^{-13}\,\text{s}$.

the reference time $t_f$ admits an alternative interpretation. Namely, the rate $\dot{h}$ at which the universe is converted from the old to the new ground state has a maximum at time $t_f$. This can be seen from the second time derivative of the fractional volume,

$$\frac{\mathrm{d}}{\mathrm{d}t}\left(\frac{\mathrm{d}h(t)}{\mathrm{d}t}\right) = \beta^2\left(1 - e^{\beta(t - t_f)}\right)\exp\left(-e^{\beta(t - t_f)} + \beta(t - t_f)\right). \tag{7.18}$$

Hence, for $t = t_f$ the above expression vanishes due to the first term in the brackets. This is also demonstrated in Fig. 11. The time[5] $t_H$ that we introduced first, is always smaller than $t_f$, i.e. $t_H < t_f$. At the time $t_H$ basically the entire universe is still in the metastable phase as can be seen in the plot. Instead, the phase transition completes very rapidly around $t_f$ within a duration set by $\beta^{-1}$. To draw the plot in Fig. 11 we use the realistic values of $T_H = 105.2\,\text{GeV}$ and $T_f = 104.9\,\text{GeV}$ for the temperatures and $\beta = 8950H$, and $v_w = 0.9$. This implies that $t_H = 2.66 \cdot 10^{-10}\,\text{s}$ and $t_f = 2.72 \cdot 10^{-10}\,\text{s}$ such that $t_f - t_H \sim 10^{-13}\,\text{s}$. Shortly after the nucleation rate $\dot{h}$ is at its maximum at $t_f$, essentially the entire universe is in the stable ground state.

The bubbles can nucleate only in the metastable phase. Hence, using Eq. (7.13) we can

---

[5]In order to find a relation between $t_H$ and $t_f$, we note that Eq. (7.13) reads $\Gamma_H = \Gamma_f \exp\left[\beta(t_H - t_f)\right]$. Using Eqs. (7.10) and (7.16), and solving for $t_H$ yields

$$t_H = t_f + \beta^{-1}\ln\left(8\pi v_w^3 \frac{H^4}{\beta^4}\right).$$

The logarithm evaluates to negative values (much) larger than unity. Therefore, the time between $t_H$ and $t_f$ is some orders of magnitude larger than $\beta^{-1}$.

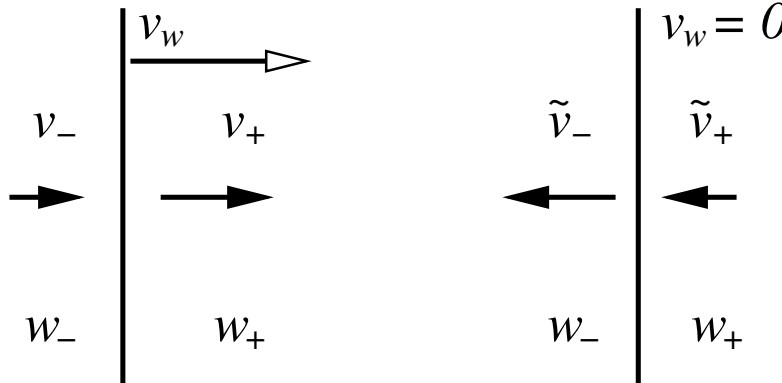

Figure 12: Fluid velocities right ahead of and behind the wall in the case of a supersonic deflagration. The velocities just ahead the wall are denoted by + while the velocities just behind the wall are denoted by a − subscript. The left panel shows the velocities in the universe rest frame in which the wall moves with the speed $v_w$. The right panel shows the velocities in the wall rest frame where $v_w = 0$ by definition and we indicate the fluid velocities in this frame by a tilde, $\tilde{v}_{\pm}$. The enthalpy is denoted by $w$. Figures taken from Ref. [66].

deduce that the bubble number density $n_{\text{bubble}}(t)$ in the universe is given by

$$n_{\text{bubble}}(t) = \int^t dt' \frac{\Gamma(t')}{\mathcal{V}} h(t') = \frac{\Gamma_f}{\mathcal{V}} \int^t dt' e^{\beta(t'-t_f)} h(t') \tag{7.19}$$

$$= -\frac{\Gamma_f}{\mathcal{V}} \frac{1}{\beta} \int^t dt' \frac{dh(t')}{dt'} = \frac{\Gamma_f}{\mathcal{V}} \frac{1}{\beta} (1 - h(t)), \tag{7.20}$$

where we used Eq. (7.17) in the second line. At late times when $t \to \infty$ and upon using Eq. (7.16), this expression simplifies to

$$n_{\text{bubble}}(\infty) = \frac{\Gamma_f}{\mathcal{V}} \frac{1}{\beta} = \frac{\beta^3}{8\pi v_w^3} \equiv \frac{1}{R_*^3}, \tag{7.21}$$

which is the final bubble density, and defines the mean bubble centre separation $R_* = (n_{\text{bubble}}(\infty))^{-1/3}$. As we will see later, the mean bubble centre separation is linked to the position of the peak of the gravitational wave power spectrum produced by the first-order phase transition.

Note that for deflagrations, one should really take into account the heating of the fluid in front of the bubble wall, which will reduced the net nucleation rate. In the extreme case of a slow wall and large latent heat, nucleation can stop altogether, and the interior of the bubbles can reheat to the critical temperature. In this case, the universe is in a mixed phase, and the transition takes about a Hubble time to complete [67].

## 7.2 Relativistic combustion

Now we will assume that the wall velocity $v_w$ is known, and discuss solutions to the hydrodynamic equations following Refs. [66, 68]. Let $u^\mu$ be again the 4-velocity of the fluid and we introduce the enthalpy $w = e + p$ of the fluid. First, we work in the wall rest frame where $v_w = 0$, see Fig. 12, and introduce a subscript $\pm$ to distinguish between quantities evaluated just ahead of the wall (+) and just behind the wall (−). From the conservation of energy-momentum at the wall it follows that

$$w_- \tilde{\gamma}_-^2 \tilde{v}_- = w_+ \tilde{\gamma}_+^2 \tilde{v}_+, \qquad w_- \tilde{\gamma}_-^2 \tilde{v}_-^2 + p_- = w_+ \tilde{\gamma}_+^2 \tilde{v}_+^2 + p_+, \tag{7.22}$$

where $\tilde{v}_\pm$ are the fluid velocities in the rest frame of the wall and $\tilde{\gamma}_\pm = (1 - \tilde{v}_\pm^2)^{-1/2}$. The enthalpy density $w$ and pressure $p$ are scalars, and so they are the same in the wall frame and the universe rest frame. These are the bubble wall junction conditions. They can be rearranged to give

$$\tilde{v}_+\tilde{v}_- = \frac{p_+ - p_-}{e_+ - e_-}, \qquad \frac{\tilde{v}_+}{\tilde{v}_-} = \frac{e_- + p_+}{e_+ + p_-}. \tag{7.23}$$

For convenience, let us introduce the trace anomaly

$$\theta = \frac{1}{4}(e - 3p), \tag{7.24}$$

which is proportional to the trace of the energy-momentum tensor. The trace anomaly should vanish for an ultra-relativistic equation of state. We denote the difference of the trace anomaly just ahead and behind the wall by $\Delta\theta = \theta_+ - \theta_-$. From here, one defines the transition strength $\alpha_+$ and the enthalpy ratio $r$ as

$$\alpha_+ = \frac{4\Delta\theta}{3w_+}, \qquad r = \frac{w_+}{w_-}. \tag{7.25}$$

In terms of these quantities, Eq. (7.23) can be rearranged as

$$\tilde{v}_+\tilde{v}_- = \frac{1 - (1 - 3\alpha_+)r}{3 - 3(1 + \alpha_+)r}, \qquad \frac{\tilde{v}_+}{\tilde{v}_-} = \frac{3 + (1 - 3\alpha_+)r}{1 + 3(1 + \alpha_+)r}. \tag{7.26}$$

These equations can be solved for $\tilde{v}_+$ in terms of $\tilde{v}_-$ (or vice versa). The result is

$$\tilde{v}_+ = \frac{1}{1 + \alpha_+}\left(\frac{\tilde{v}_-}{2} + \frac{1}{6\tilde{v}_-} \pm \sqrt{\left(\frac{\tilde{v}_-}{2} - \frac{1}{6\tilde{v}_-}\right)^2 + \frac{2}{3}\alpha_+ + \alpha_+^2}\right), \tag{7.27}$$

which depends only on $\tilde{v}_-$ and $\alpha_+$ but not on $r$. As we explain in more detail later, the upper sign is to be taken for $\tilde{v}_- > 1/\sqrt{3}$, while the lower sign is to be taken if $\tilde{v}_- < 1/\sqrt{3}$ in order to ensure the solution to be physical. This implies that both velocities are either subsonic (deflagrations) or supersonic (detonations). Further, $\tilde{v}_+$ must be positive because the fluid must flow through the wall from the outside to the inside of the bubble. This requires $\alpha_+ < 1/3$. In Fig. 13, $\tilde{v}_+$ as a function of $\tilde{v}_-$ is shown for different values of $\alpha_+$.

So far, from the conservation of energy-momentum, we obtained a relation between the fluid velocities ahead and behind the wall which also allows one to compute the enthalpies $w$. We can obtain two more independent equations by projecting the conservation equation onto the fluid 4-velocity $u^\mu = \gamma(1, \vec{v})^\mu$ and the space-time orthonormal vector $\bar{u}^\mu = \gamma(v, \vec{v}/v)^\mu$. These vectors satisfy $\bar{u}_\mu\bar{u}^\mu = 1$ and $u_\mu\bar{u}^\mu = 0$. Projecting the conservation equation yields

$$0 = u_\mu \partial_\nu T^{\mu\nu} = -\partial_\mu(wu^\mu) + u^\mu\partial_\mu p, \tag{7.28}$$
$$0 = \bar{u}_\mu \partial_\nu T^{\mu\nu} = w\bar{u}^\nu u^\mu\partial_\mu u_\nu + \bar{u}^\mu\partial_\mu p, \tag{7.29}$$

which are referred to as continuity equations.

In order to simplify these equations, we assume that the bubbles are spherically symmetric. The radius of the bubble is given by $R = v_w t$ where we set the nucleation time to $t' = 0$. Since there is no length scale involved in the problem, the differential equations should exhibit similarity solutions that depend on the dimensionless coordinate $\xi = r/t$ only. We can write the fluid velocity as $\vec{v} = v(r, t)\vec{r} = v(\xi)\vec{r}$, where $\vec{r}$ is a unit radial vector. The continuity equations can be rearranged to

$$\frac{dv}{d\xi} = \frac{2v(1 - v^2)}{\xi(1 - \xi v)}\left(\frac{\mu^2}{c_s^2} - 1\right)^{-1}, \tag{7.30}$$

$$\frac{dw}{d\xi} = w\left(1 + \frac{1}{c_s^2}\right)\gamma^2\mu\frac{dv}{d\xi}. \tag{7.31}$$

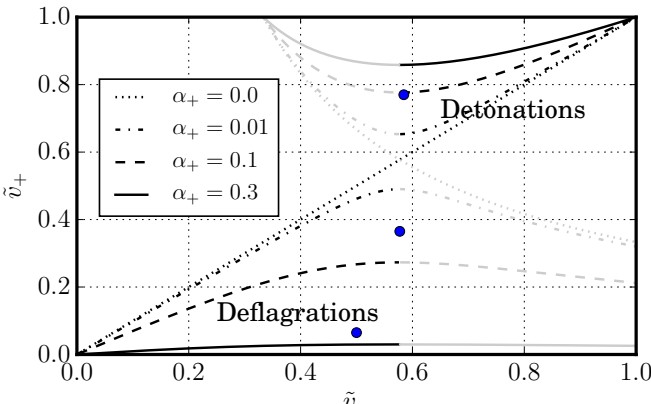

Figure 13: The fluid velocity just ahead the wall $\tilde{v}_+$ as a function of the fluid velocity just behind the wall $\tilde{v}_-$ in the rest frame of the wall for different transitions strength parameters $\alpha_+$. For physical solutions to exist, both velocities have to be subsonic (deflagrations) or supersonic (detonations) as indicated by the black lines. Non-physical solutions are indicated by the gray lines. The fluid velocity has an extremum at $\tilde{v}_+ = c_s$ with the speed of sound set to $c_s = 1/\sqrt{3}$ in this plot. The velocity profiles at the blue dots are presented in Fig. 15. Figure taken from Ref. [66]

Here, $c_s^2 = dp/de$ is the speed of sound and

$$\mu = \frac{\xi - v}{1 - \xi v} \tag{7.32}$$

is the fluid velocity at $\xi$ in a frame that is moving outward at speed $\xi$. Solving these equations requires numerical techniques.

In general, the speed of sound $c_s$ depends on the temperature which changes across the wall. The situation simplifies if one assumes an ultrarelativistic equation-of-state in both phases such that $c_s^2 = 1/3$ everywhere. An example of such an equation of state is the "bag" model, where

$$p_s = a_s T^4 - V_s, \quad p_b = a_b T^4, \tag{7.33}$$

where $a_s$, $a_b$ and $V_s$ are positive constants with $a_s > a_b$. The subscripts s and b refer to the symmetric (i.e., where $\phi = 0$) and broken (i.e., where $\phi = \phi_b$) phases respectively. In this case, the fluid speed equation can be integrated separately. More realistic equations of state have also recently been considered in Ref. [69].

Solutions are obtained by integrating the continuity equations starting at the position of the wall, $\xi_w = v_w$, while the wall is assumed to be infinitesimally thin. The boundary conditions at the wall read $v \to v_\pm = \mu(\xi_w, \tilde{v}_\pm)$ as $\xi \to \xi_w^\pm$ where $\xi_w^\pm = v_w \pm \delta$ and $\delta$ infinitesimally small. On the other hand, the fluid velocity must vanish, $v = 0$, at the center of the bubble $\xi = 0$ due to spherical symmetry and at $\xi = 1$ due to causality because we assume the fluid to be undisturbed until a signal from the wall arrives.

The solutions can be classified by how the boundary conditions are satisfied. Specifically, there are only two possibilities to smoothly approach $v = 0$. Either one starts with $v = 0$ or one starts in the region $\xi > c_s$ and $\mu(\xi, v) > c_s$ implying that $dv/d\xi > 0$ and then integrating backwards in $\xi$. The only other way to meet the boundary conditions is by a discontinuity, i.e., by a shock. Hence, we can identify the following three classes of solutions:

- *Subsonic deflagrations:* In a subsonic deflagration the wall moves at a subsonic speed, $v_w < 1/\sqrt{3}$, and the fluid is at rest everywhere inside the bubble, $v_- = 0$. In the wall rest

frame we thus find $\tilde{v}_- = v_w$. The fluid velocity just ahead the wall in the universe frame is $v_+ = \mu(\tilde{v}_+(\alpha_+, v_w), \xi_w)$. In order to ensure $v_+ > 0$ one has to choose the negative sign in Eq. (7.27) as anticipated. Further one finds that ahead of the wall the fluid velocity decreases until a shock occurs outside which the fluid is at rest. This situation is depicted schematically in the left panel of Fig. 14. The velocity profile is shown in the upper left panel of Fig. 15.

- *Detonations:* A detonation is characterized by a fluid exit speed in the wall frame $\tilde{v}_- > 1/\sqrt{3}$, with the fluid being at rest everywhere outside the bubble, $v_+ = 0$. In the wall rest frame we thus have $\tilde{v}_+ = v_w$. The condition on $\tilde{v}_-$ means there is a minimum for $\tilde{v}_+$ called the Chapman-Jouguet speed $v_{CJ}$. It is given by

$$v_{CJ}(\alpha_+) = \frac{1}{\sqrt{3}} \left( \frac{1 + \sqrt{\alpha_+ + 3\alpha_+^2}}{1 + \alpha_+} \right). \tag{7.34}$$

In order to ensure that $\tilde{v}_-(\alpha_+, v_w) > c_s$ and hence $dv/d\xi > 0$ the positive sign has to be taken in Eq. (7.27) as mentioned before. As result, the fluid velocity smoothly decreases as one moves further inside the bubble until it is at rest at $\xi = c_s$. The schematic visualization can be found in the right panel of Fig. 14 while the velocity profile is shown in the upper right panel of Fig. 15.

- *Supersonic deflagrations (hybrids):* There is a hybrid between the two aforementioned classes of solutions with the wall moving at supersonic speed, $v_w > 1/\sqrt{3}$, which occurs for $\tilde{v}_- = 1/\sqrt{3}$. A physical solution for $\tilde{v}_+(\alpha_+, 1/\sqrt{3})$ exists provided that it is larger than the wall speed to ensure a positive $v_+$. In front of the wall, the fluid behaves in exactly the same manner as for a subsonic deflagration. The hybrid is schematically shown in the middle panel of Fig. 14 while the velocity profile is shown in the upper middle panel of Fig. 15.

In the lower panels of Fig. 15 we present the enthalpy profiles for all three classes. Note that the solutions are found once $\alpha_+$ is given, which requires knowing the energy density and pressure just in front of the wall. However, one normally specifies these quantities far in front of the wall, where the undisturbed plasma is at the nucleation temperature. For detonations, this presents no problem, but for deflagrations shooting method is needed to find the correct value of $\alpha_+$ which matches to the asymptotic energy and pressure.

For more details on these solutions we refer to Refs. [66, 68–70].

## 7.3 Energy redistribution

Roughly speaking, the expanding bubble converts potential energy of the scalar field into kinetic energy and heat. The kinetic energy fraction of a single bubble is a reasonable estimate of the kinetic energy fraction of the entire fluid flow [66, 71]. The kinetic energy fraction of a single bubble can then be used to estimate the power of the gravitational wave signal. Hence, making the above statement quantitatively precise is crucial.

To remind ourselves, the spatial components of fluid energy-momentum tensor are given by

$$T^i_j = w\gamma^2 v^i v_j + p\delta^i_j, \tag{7.35}$$

with $\vec{v}$ the fluid velocity, $w = e + p$ the enthalpy, $e$ the total energy and $p$ the pressure of the fluid. The kinetic energy of the fluid can be obtained as the trace of the energy-momentum tensor minus the trace in the rest frame of the fluid, and integrating over space. The kinetic energy fraction is thus given by

$$K = \frac{1}{\mathcal{V}\bar{e}} \int d^3x \, w\gamma^2 v^2, \tag{7.36}$$

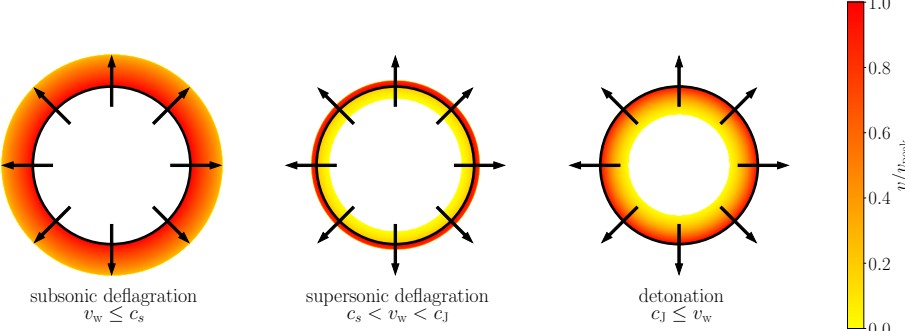

Figure 14: Sketch of the three different cases of relativistic combustion. A subsonic deflagration (left) occurs detonation when the fluid is at rest inside the bubble and the wall moves at subsonic speed. The opposite case is a detonation (right) where the fluid outside the bubble is at rest and the wall moves at supersonic speed. In the hybrid case (mid) the wall speed is supersonic and the fluid is moving both ahead and behind the wall. Credit: D. Cutting.

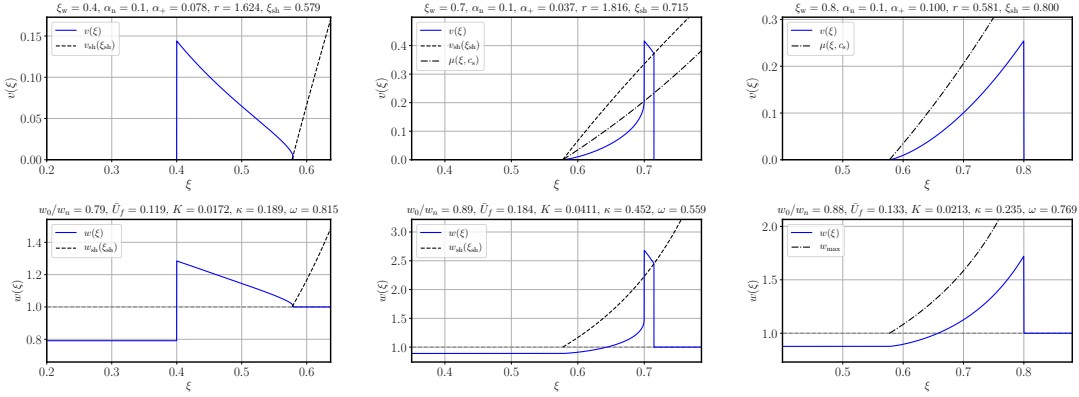

Figure 15: The fluid velocity $v$ and the enthalphy $w$ as a functions of the dimensionless coordinate $\xi = r/t$ for the three classes of solutions is depicted. The wall is located at $\xi_\mathrm{w}$. The left panel shows the velocity profile for a subsonic deflagration, where the fluid velocity smoothly decreases from the wall until it reaches a shock ahead of which the fluid is at rest. The middle panel shows supersonic deflagration (hybrid). The right panel shows a detonation, where the velocity smoothly increases until the location of the wall, while the fluid is at rest outside the bubble. The three cases are represented by the blue dots in Fig. 13. Figures taken from Ref. [66].

where $\mathcal{V}$ is the averaging volume, and $\bar{e}$ is the mean energy density.

The kinetic energy fraction of a single bubble describes how much of the initially available energy contained in the bubble is converted into kinetic energy, which can

$$K_1 = \frac{3}{\xi_\mathrm{w}^3 e_\mathrm{s}} \int \mathrm{d}\xi \, \xi^2 w \gamma^2 v^2 \,, \tag{7.37}$$

where $e_\mathrm{s}$ is the mean energy density in the symmetric phase. Conservation of energy, and the rapidity of the transition compared with the Hubble rate, implies that the mean energy density in the broken phase $e_\mathrm{b} = e_\mathrm{s}$. Our estimate of the kinetic energy fraction in the broken phase is therefore $K = K_1$.

It is also useful to define the enthalpy-weighted root-mean-square 4-velocity of the fluid

$\bar{U}_{\rm f}$, through

$$\bar{U}_{\rm f}^2 = \frac{3}{\xi_{\rm w}^3 \bar{w}} \int {\rm d}\xi \, \xi^2 w \gamma^2 v^2 \,, \tag{7.38}$$

which can be related to the kinetic energy fraction via

$$K = \Gamma \bar{U}_{\rm f}^2 \,, \tag{7.39}$$

where $\Gamma = \bar{w}/\bar{e}$ is the adiabatic index of the fluid in the broken phase.

In the previous section we introduced the transition strength parameter $\alpha_+$, which is defined in terms the enthalpy and the trace anomalies just ahead and behind the wall. These quantities are perturbative by definition and thus $\alpha_+$ is difficult to compute in practice. Instead, it would be nice to have a quantity that measures the transition strength in terms of background variables only. As before, we denote variables that correspond to the symmetric phase (i.e., where $\phi = 0$) with a subscript s while variables that correspond the broken phase (i.e., where $\phi = \phi_{\rm b}$) with a subscript b. In terms of these background variables, one defines the transition strength parameter as

$$\alpha_{\rm n} = \frac{4}{3} \frac{\Delta \theta}{w_{\rm s}} \bigg|_{T = T_{\rm n}} \,, \tag{7.40}$$

where now $\Delta \theta = \theta_{\rm s} - \theta_{\rm b}$. The two definitions of the transition strength coincide only in the case of detonations within the bag model. The transition strength parameter is defined at the nucleation temperature $T_{\rm n}$. Note that in the literature there exist various definitions of the transition strength parameter $\alpha$ that are related in nontrivial and model-dependent ways.

It is convenient to define the *efficiency factor* $\kappa$ that quantifies how much of the available energy is converted into kinetic energy. The available energy is determined by the trace anomaly $\theta$ introduced in Eq. (7.24). It is related to the potential energy of the scalar field $V_T(\phi)$ as

$$\theta = V_T(\phi) - \frac{1}{4} T \frac{\partial V_T}{\partial T} \,. \tag{7.41}$$

This implies that not all of the potential energy can be converted into kinetic energy and heat. Therefore, one defines the efficiency factor as[6]

$$\kappa = \frac{3}{\xi_{\rm w}^3 \Delta \theta} \int {\rm d}\xi \, \xi^2 w \gamma^2 v^2 \,, \tag{7.42}$$

where $\Delta \theta$ is again the difference of the trace anomaly between the symmetric and broken phase. Consequently, efficiency factor, transition strength and kinetic energy fraction are related as

$$K = \frac{\kappa \alpha_{\rm n}}{1 + \alpha_{\rm n} + \delta_{\rm n}} \,, \tag{7.43}$$

where $\delta_{\rm n} = 4\theta_{\rm b}/(3w_{\rm s})$. The efficiency factor can be computed numerically as a function of $\alpha_{\rm n}$ and $\xi_{\rm w}$ as in Ref. [68]. To give an explicit example, in the case of detonations and using the bag model, where $\theta_{\rm b} = 0$, they find the approximate relation

$$\kappa \simeq \frac{\alpha_{\rm n}}{0.73 + 0.083 \sqrt{\alpha_{\rm n}} + \alpha_{\rm n}} \tag{7.44}$$

for $\xi_{\rm w} \to 1$. For $\alpha_{\rm n} < 1$, one can generally take $\kappa \sim \alpha_{\rm n}$, except for low wall speeds, and wall speeds near the Chapman-Jouguet speed, where the parametric dependence is closer to $\kappa \sim \sqrt{\alpha_{\rm n}}$ [68]. In addition, relations between the transition strength and the inverse duration of phase transition were recently established for a variety of BSMs in Ref. [72, 73].

---

[6]In the bag model, it is assumed that $\theta_{\rm b} = 0$ such that $\Delta \theta$ coincides with the so-called bag constant $\epsilon$.

### 7.4 Sound waves

Radial perturbations of the radial velocity field are longitudinal in character, and so we can think of the bubble expansion as an explosion, generating a compression wave which, once the bubbles have disappeared, propagate through the fluid as sound waves. We conclude this section by studying sound waves in a relativistic fluid.

Let us work in the plane wall approximation and let the $z$-direction be orthogonal to the wall. Then, the wall moves only in the $z$-direction and also the fluid is perturbed in the $z$-direction only. Let us write the perturbed energy density as $e = \bar{e} + \delta e$ and the perturbed pressure as $p = \bar{p} + \delta p$ with $\{\delta e, \delta p, v^z\} \ll 1$ with $v^z$ the fluid velocity perturbation. The components of the fluid energy-momentum tensor simplify to

$$T^{tt} = w\gamma^2 - p, \quad T^{tz} = w\gamma^2 v^z, \quad T^{zz} = w\gamma^2 (v^z)^2 + p, \tag{7.45}$$

while the other components vanish identically. From the $t$-component of the energy-momentum conservation equation we find

$$\partial_t \delta e + \bar{w}\partial_z v^z = 0 \tag{7.46}$$

and from the $z$-component

$$\bar{w}\partial_t v^z + \partial_z \delta p = 0. \tag{7.47}$$

Note that both $\delta e$ and $\delta p$ depend on temperature as $\delta p = (\frac{\partial p}{\partial T} / \frac{\partial e}{\partial T})\delta e = c_s^2 \delta e$. Therefore, the $t$- and $z$-components of the conservation equation can be combined to

$$(\partial_t^2 - c_s^2 \partial_z^2)v^z = 0, \quad (\partial_t^2 - c_s^2 \partial_z^2)\delta p = 0. \tag{7.48}$$

These equations describe sound waves that travel at the speed $c_s$ through the fluid. Sound waves are a collective mode of the fluid velocity $v$ and temperature $T$. They are longitudinal as the fluid velocity varies along the direction of travel.

## 8 Gravitational Waves

Gravitational waves were predicted by Einstein in 1916 [74, 75], although it took about forty years for them to be understood as physical, rather than coordinate artefacts (see e.g. [76]). According to the general theory of relativity, they are perturbations of space-time that travel at the speed of light. They are generated by an accelerating asymmetric mass distribution; more precisely, a distribution of energy-momentum with a time-dependent quadrupole moment. The strongest astrophysical sources of gravitational waves are compact binary systems: combinations of neutron stars, black holes or white dwarfs. Besides these astrophysical sources there are also cosmological sources, in particular the early Universe (see e.g. [22]). In this section we give a rough overview about the production of gravitational waves by first-order phase transitions. For a detailed discussion we refer to the review paper by the LISA working group [12]. Good textbooks are [77, 78]. Fig. 16 gives an overview over the spectrum of gravitational waves and possible sources.

### 8.1 Introduction

Gravitational waves yield expansion of spacetime in one direction and contraction in the other, in the plane perpendicular to the propagation direction. Modern detection methods are mostly based on measuring the variation of the distance between two test masses by interferometry, as sketched in Fig. 17. The first direct detection was performed in 2015 by the LIGO/VIRGO science collaborations [10].

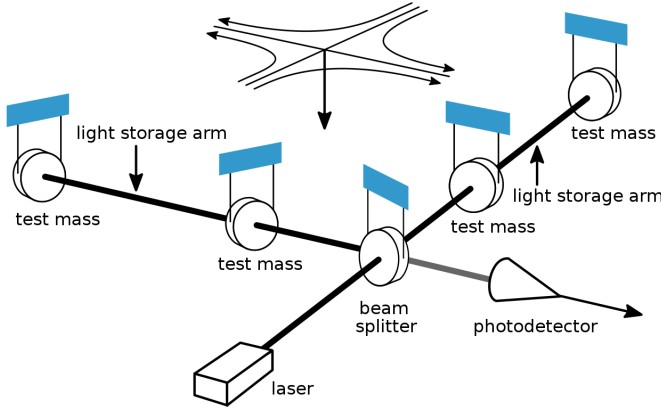

Figure 16: The spectrum of gravitational waves with possible sources and detectors. The gravitational wave spectrum from first-order phase transitions in the early universe are expected to peak in the sensitivity range of LISA. Credit: NASA.

The results of the first measurement [10] showed that two black holes with masses $M_1 = 36 \pm 6 M_\odot$ and $M_2 = 29 \pm 4 M_\odot$ produced the signal. According to numerical simulations of such an event, an energy equivalent of $M_{\mathrm{gw}} \simeq 3 M_\odot$ must have been radiated away as gravitational waves.

Figure 17: Sketch of the gravitational wave detector LIGO.

After this first direct detection, in 2017 gravitational waves of the merger of a binary neutron star system were observed [79,80]. This signal was accompanied by the detection of a gamma ray burst (GRB) which allows to put constraints on the travel speed of gravitational waves $c_{\mathrm{gw}}$. It was found that gravitational waves travel at speed of light to a large precision,

$$|c_{\mathrm{gw}}^2 - 1| \lesssim 10^{-5}\,. \tag{8.1}$$

This puts tight constraints on modifications to general relativity [81,82]. A total of 11 secure detections are reported from the first two observing runs (O1 and O2), with 56 further candidates for further analysis from O3, which ended in March 2019. Further exciting results

Table 3: Current and planned (*) ground-based gravitational wave detectors using laser interferometry.

| Name | Location | Arm length |
|---|---|---|
| GEO | Germany | 0.6km |
| (a)LIGO | USA (2) | 4km |
| (a)VIRGO | Italy | 3km |
| KAGRA | Japan | 4km |
| LIGO-India* | India | 4km |

Table 4: Overview about the minimal frequencies of gravitational waves from different potential sources in the early universe.

| Event | Time [s] | Temperature [GeV] | Frequency [Hz] |
|---|---|---|---|
| QCD phase transition | $10^{-3}$ | 0.1 | $10^{-8}$ |
| EW phase transition | $10^{-11}$ | 100 | $10^{-5}$ |
| ? | $10^{-25}$ | $10^9$ | 100 |
| End of inflation | $\geq 10^{-36}$ | $\leq 10^{16}$ | $\geq 10^8$ |

are expected from the expanding network of detectors. In Tab. 3 we summarize existing and planned ground-based interferometric gravitational wave detectors.

We are interested in gravitational waves that are produced in the early Universe. As a rough estimate of the frequency, take an event at time $t$ that produces gravitational waves. The minimal frequency is given by $f = t^{-1} \sim H$ with $H = \dot{a}/a$ the Hubble rate of that time and $a$ the scale factor. Due to cosmic expansion, the minimal frequency that we can observe today is redshifted as $f_0 = a(t)/a(t_0)f$, where $t_0$ is the time now. In Tab. 4 we summarize the minimal frequencies from different events in the early Universe.

Before going on, let us be a bit more quantitative in our outline of gravitational waves. We start by considering perturbations about Minkowski background $\eta_{\mu\nu}$ as

$$g_{\mu\nu} = \eta_{\mu\nu} + h_{\mu\nu}, \tag{8.2}$$

where $h_{\mu\nu} \ll 1$ is the metric perturbation. A gravitational wave $h_{ij}(t, \vec{x})$ is a propagating mode of the transverse ($\partial_i h_{ij} = 0$) and traceless ($h_{ii} = 0$) part of the metric perturbation, satisfying the equation

$$\ddot{h}_{ij} - \nabla^2 h_{ij} = 16\pi G T_{ij}^{\mathrm{TT}}, \tag{8.3}$$

where $T_{ij}^{\mathrm{TT}}$ is the transverse and traceless part of the energy-momentum tensor, and $G$ is Newton's constant. The constraints on $h_{ij}$ reduce the number of physical propagating modes to two, called 'plus' (+) and 'cross' (×) polarisations.

Provided their amplitude is small, gravitational waves can themselves be a source of energy-momentum, described by the tensor [78]

$$T_{\mu\nu}^{\mathrm{gw}} = \frac{1}{32\pi G} \langle \partial_\mu h_{ij} \partial_\nu h_{ij} \rangle. \tag{8.4}$$

We aim at computing the power spectrum of the energy density of gravitational waves. From the $tt$-component of the energy-momentum tensor, we find the energy density,

$$\rho_{\mathrm{gw}} = \frac{1}{32\pi G} \langle \dot{h}_{ij}^2 \rangle. \tag{8.5}$$

After a spatial Fourier transform of the metric, one can consider the contribution of a frequency interval $df$ to the total energy density, where the frequency of the gravitational wave $f$ is

related to the wavenumber of the Fourier transform $k$ by $f = k/2\pi$. More often, one studies the energy density per logarithmic frequency interval $\mathrm{d}\ln f$, which can be written

$$\frac{\mathrm{d}\rho_{\mathrm{gw}}}{\mathrm{d}\ln f} = \frac{\pi}{4G}f^3 S_h(f), \tag{8.6}$$

which can be treated as a definition of the one-sided strain power spectral density $S_h(f)$, defined for positive frequencies, $f > 0$. From here, one can indicate the amplitude of a gravitational wave spectrum at frequency $f$ by the root power spectral density $h(f) = \sqrt{S_h(f)}$, which has dimension $\sqrt{\mathrm{Hz}^{-1}}$, or the (dimensionless) characteristic strain $h_c(f) = \sqrt{f\,S_h(f)}$.

A convenient measure in cosmology is the fractional density in gravitational waves

$$\Omega_{\mathrm{gw}} = \frac{\rho_{\mathrm{gw}}}{\rho_{\mathrm{tot}}}, \tag{8.7}$$

where $\rho_{\mathrm{tot}}$ is the total energy density of the universe, related to the Hubble rate $H$ by the Friedmann equation,

$$H^2 = \frac{8\pi G \rho_{\mathrm{tot}}}{3}. \tag{8.8}$$

With these quantities, one can define the fractional gravitational wave energy density per logarithmic frequency,

$$\frac{\mathrm{d}\Omega_{\mathrm{gw}}}{\mathrm{d}\ln f} = \frac{1}{\rho_{\mathrm{tot}}}\frac{\mathrm{d}\rho_{\mathrm{gw}}}{\mathrm{d}\ln f} = \frac{2\pi^2}{3H^2}f^3 S_h(f) \tag{8.9}$$

often referred to as the gravitational wave (power) spectrum. The characteristic strain and the fractional density in gravitational waves are related through

$$h_c(f) = \frac{H_0}{f}\sqrt{\frac{3}{2\pi^2}\frac{\mathrm{d}\Omega_{\mathrm{gw}}}{\mathrm{d}\ln f}}. \tag{8.10}$$

Going from Minkowski space to the Friedmann-Lemaître-Robertson-Walker metric describing an expanding universe, one finds that the gravitational waves behave just like electromagnetic radiation: the frequency is redshifted, and the energy density decreases as the fourth power of the scale factor [77].

The metric perturbations $h_{ij}$ cause changes in lengths in directions perpendicular to the propagation direction of the wave, which can be measured by interferometry. Let $l_i$ and $m_i$ be components of unit vectors along the arms of an interferometer located at $\vec{x}$. The strain, or relative change in length, in the detector, in the limit that the arm length is much less than the wavelength of the gravitational wave, is

$$h(t) = \frac{1}{2}h_{ij}(t,\vec{x})(l_i l_j - m_i m_j), \tag{8.11}$$

whose Fourier transform is

$$\tilde{h}(f) = \frac{1}{2}\int_{-\infty}^{\infty}\mathrm{d}t\, e^{-2\pi i f t}h_{ij}(l_i l_j - m_i m_j). \tag{8.12}$$

We define the one-sided strain power spectral density of the signal in the interferometer $S_h^I(f)$ as

$$\langle\tilde{h}(f)\tilde{h}^*(f')\rangle = \frac{1}{2}S_h^I(f)\delta(f - f'). \tag{8.13}$$

This is not the same quantity as the strain power spectral density in a gravitational wave. In general they are related by a response function $\mathcal{R}^I(f)$,

$$S_h^I(f) = \mathcal{R}^I(f)S_h(f). \tag{8.14}$$

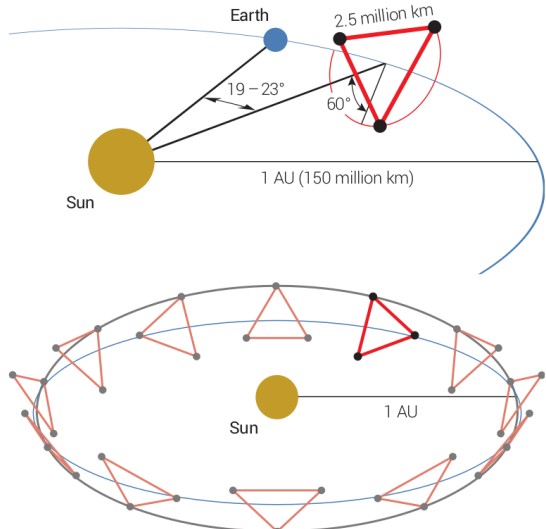

Figure 18: Visualization of the orbit of LISA. LISA is a space-based laser interferometer designed to measure gravitational waves with frequencies of about $10^{-2}$ Hz. The interferometer consists of three satellites that are orbiting the Earth in a triangle. Figure taken from Ref. [9].

For an interferometer with perpendicular arms, whose arm length is much less than the wavelength so that $fL \to 0$, $\mathcal{R}^I(f) \to 1/5$ [83].

For gravitational waves sourced by phase transitions, the relevant mission is Laser Interferometer Space Antenna (LISA). LISA is a space-based interferometric gravitational wave detector that works with three satellites orbiting the Earth, see Fig. 18. These are equipped with lasers and photodetectors such as to detect small changes in the separation of the satellites by measuring the time delays in signals sent between them, through the technique of time delay interferometry [84–86]. The interferometer arms have a length of 2.5 Gm such that LISA is most sensitive at frequencies in the range $10^{-3}$ – $10^{-2}$ Hz. The target sources are binary white dwarfs, merging supermassive black holes at the centres of distant galaxies, and early-universe physics at the TeV-scale such as first-order phase transitions. The planned launch year is 2034. In Fig. 19 we show the projected sensitivity of LISA and the amplitudes of expected astrophysical sources. In addition we have added the signal from a first-order phase transition, which we discuss further below. As can be seen, LISA is expected to measure gravitational waves coming from phase transitions, but other sources produce a signal in the same frequency interval as well. For details on LISA, the expected sources of gravitational waves, and its relevance for probing fundamental physics we recommend Refs. [9, 12, 87]. Other planned space-based gravitational missions include DECIGO [88], Taiji [89] and TianQin [90].

## 8.2 GWs from first-order phase transitions

In this section we discuss the expected gravitational waves signal produced by a first-order phase transition. The details and derivation are quite involved and require numerical techniques which is beyond the scope of this course. Instead, we explain the key quantities that can be inferred from the gravitational wave power spectrum and their relation to the results of the previous sections. For a review we refer to Ref. [12] that discuss the predicted power spectra for various BSMs.

The general picture is the following. Initially, when temperatures are high, $T \gg T_{\rm c}$, the thermal Higgs potential (free energy) has a minimum only at $\phi = 0$. We recall that minima

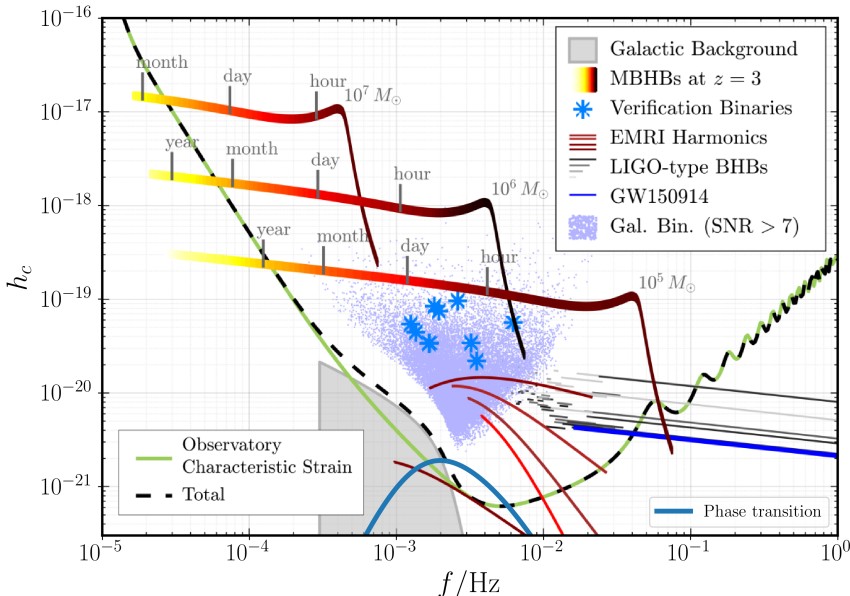

Figure 19: The expected sensitivity of LISA and possible astrophysical sources of gravitational waves. The phase transition signal, indicated by the blue line, is predicted for a transition with nucleation temperature $T_n = 500$ GeV, strength parameter $\alpha = 0.3$, transition rate to Hubble rate ratio $\beta/H_n = 100$, and wall speed $v_w = 0.4$. Background figure taken from [9].

of the free energy correspond to equilibrium states. Therefore, the Higgs is everywhere in the symmetric phase with thermal fluctuations around the minimum. As the temperature drops, the thermal Higgs potential develops a second minimum and the minima are separated by a potential barrier. At a critical temperature $T = T_c$, both minima are degenerate. Below the critical temperature, it is thermodynamically preferred for the Higgs to occupy the new minimum. However, it has to cross the potential barrier. This it can do either by quantum tunneling [49, 91] or by thermal fluctuations [50]. Here, we will study the thermal case.

Thermal fluctuations can drive the Higgs into the symmetry-breaking phase in limited regions of space. The most probable shape of the regions is a spherical bubble, inside which the Higgs is already in the new phase, while outside the bubbles the Higgs is still in the metastable phase. Large enough bubbles expand into the fluid such that the rest of the universe enters the symmetry-broken phase.

We have already discussed the rate at which bubbles appear, and how they expand, in sections 6 and 7. Once the bubbles start colliding, we must rely on a mixture of numerical simulations (see e.g. Ref. [71]) and modelling to describe the motion of the fluid and the production of gravitational waves.

We recall that near equilibrium, the system can be described as an ultrarelativistic fluid coupled to a Higgs field, according to equations (5.26) and (5.24). For concreteness, we state the equations of motion as used in Ref. [71] ready for numerical simulations. As before, the fluid four-velocity is given by $u^\mu = \gamma(1, \vec{v})^\mu$. The Higgs field satisfies the relativistic equation

$$-\ddot{\phi} + \nabla^2 \phi - V' = \eta\gamma(\dot{\phi} + v^i\partial_i\phi), \tag{8.15}$$

with $\eta$ the Higgs-fluid coupling and $V' = \partial_\phi V$, c.f. Eq. (7.2). Further the relativistic fluid

equations read

$$\dot{E} + \partial_i(Ev^i) + p(\dot{\gamma} + \partial_i(\gamma v^i)) - V'\gamma(\dot{\phi} + v^i\partial_i\phi) = \eta^2\gamma^2(\dot{\phi} + v^i\partial_i\phi)^2$$
$$\dot{Z}_i + \partial_j(Z_i v^j) + \partial_i p + V'\partial_i\phi = -\eta\gamma(\dot{\phi} + v^j\partial_j\phi)\partial_i\phi,$$

which follow from Eq. (5.24) together with Eq. (5.25). Here, $E = \gamma e$ is the fluid energy density and $Z_i = \gamma(e + p)u_i$ the fluid momentum density with $u_i = \gamma v^i$. The uncertainty in the precise form of $\eta$ turns out not to matter much: different choices affect only the precise profile of the domain wall as it moves through the fluid, and the hydrodynamics ensures that the fluid flows around expanding bubbles are the same for a given wall speed $v_w$ and phase transition strength $\alpha_n$.

Finally, the metric perturbation is described by Eq. (8.3). In practice, one evolves an unconstrained symmetric tensor $u_{ij}$ with the tensor source

$$\Pi_{ij} = (e + p)\gamma^2 v_i v_j + \partial_i\phi\partial_j\phi, \tag{8.16}$$

and projects out the transverse traceless part $h_{ij}(t, \vec{k})$ from the Fourier transform $u_{kl}(t, \vec{k})$ with the appropriate projector. In the bubbles of a thermal transition, the scalar field part of the source is confined to a microscopically thin wall, while the fluid source is distributed over a significant volume. The ratio of the energies can be estimated as $\ell_w/R$, where $\ell_w$ is the wall thickness, and $R$ is the bubble radius. The wall thickness $\ell_w$ is very roughly the inverse temperature, and given that the total energy density is $e \sim T^4$, the ratio can be estimated as

$$\frac{\ell_w}{R} \sim \frac{\sqrt{G}T}{HR},$$

where we have used $H \sim \sqrt{G}T^2$ as a consequence of Friedmann's equation. At electroweak temperatures $T \sim 100$ GeV, we have $\sqrt{G}T \sim 10^{-17}$, while bubbles typically grow to a few orders of magnitude of the Hubble length. Hence, we find that $\ell_w/R \ll 1$ and the energy-momentum of the scalar field is generally negligible.

An exception is if the bubble wall is very weakly coupled to the fluid and continues to accelerate rather than reaching a terminal velocity, or there is a lot of supercooling and the fluid energy density becomes negligible compared with the potential energy in the scalar field (see [92] for a recent study). In this case, one can study the scalar field only [55, 93].

Based on simulations and modelling of thermal transitions, it is possible to identify three stages of gravitational wave production.

1. Initially the bubbles of the stable phase collide and merge. This stage is of short duration and subdominant compared to the subsequent stages of gravitational wave production, unless the bubbles grow as large as the Hubble length. We will denote the contribution following from the collision of bubbles $\Omega_{bc}$.

2. After the bubbles have collided and merged, the shells of fluid kinetic energy continue to expand into the plasma as sound waves. These different waves overlap and source gravitational waves. The power spectrum sourced by this 'acoustic' stage is denoted $\Omega_{sw}$.

3. The last stage is the so-called turbulent phase, where the intrinsic non-linearity in the fluid equations becomes important. Through the non-linearities, the previous phases might produce vorticity and turbulence, and the sound waves eventually develop shocks. The spectrum of this phase is labelled by $\Omega_{tu}$.

These different sources are relevant on different length scales and at different time scales, and the total power spectrum can be approximately written as the sum

$$\Omega_{gw} = \Omega_{bc} + \Omega_{sw} + \Omega_{tu}. \tag{8.17}$$

In general, $\Omega_{\mathrm{sw}}$ is thought to yield the dominant contribution, unless the transition is very strong, or the bubbles are as large as the cosmological horizon.

The key quantities which determine these power spectra are the following:

- $T_{\mathrm{n}}$, the cosmological phase transition nucleation temperature;

- $\alpha_{\mathrm{n}}$, the phase transition strength parameter at the nucleation temperature, which is related to the scalar potential energy released during the phase transition and given in Eq. (7.40);

- $v_{\mathrm{w}}$, the bubble wall speed.

- $\beta$, the transition rate parameter (7.12), which can be thought of as the inverse phase transition duration, and in combination with $v_{\mathrm{w}}$ determines the mean bubble centre separation $R_*$ through Eq. (7.21).

The belief is that *only* these four parameters determine the power spectrum, and therefore that the gravitational wave power spectrum is potentially a precise probe of these quantities, which are in principle computable from the Lagrangian of a specific theory.[7] Hence, there is great excitement about the potential of LISA to probe physics beyond the Standard Model [8].

One can estimate that the gravitational wave density parameter after a phase transition is [71, 96]

$$\Omega_{\mathrm{gw}} \sim (H_{\mathrm{n}}\tau_{\mathrm{v}})(H_{\mathrm{n}}\tau_{\mathrm{ac}})K^2, \tag{8.18}$$

where $\tau_{\mathrm{v}}$ is the lifetime of the shear stresses, $\tau_{\mathrm{ac}}$ is the autocorrelation time of the fluid flow, and $K$ is the kinetic energy fraction, defined in Eq. (7.36). The timescales depend on the properties of the fluid flow through its characteristic length scale $L_{\mathrm{f}}$ and a characteristic flow speed $V_{\mathrm{f}} \simeq \sqrt{K}$, which both ultimately depend on the parameters $R_*$, $\alpha_{\mathrm{n}}$ and $v_{\mathrm{w}}$. The Hubble time $H_{\mathrm{n}}^{-1}$ is a maximum time scale, as the expansion of the universe reduces the factor $H_{\mathrm{n}}^2$ in (8.18) and therefore the effective power of the source. Initially, one can estimate the characteristic length scale as $L_{\mathrm{f}} \sim R_*$. The kinetic energy fraction $K(v_{\mathrm{w}}, \alpha_{\mathrm{n}})$ of an individual bubble is a good estimate of the global kinetic energy fraction [71, 96], unless the transition is strong and proceeds by deflagrations [97]. In that case, there is significant suppression of the fluid kinetic energy, which is associated with the slowing of the bubble walls when they encounter a hot region around another bubble.

For sound waves the characteristic speed is the sound speed $c_{\mathrm{s}} \simeq 1/\sqrt{3}$, and the autocorrelation time is therefore $\tau_{\mathrm{ac}} \sim R_*/c_{\mathrm{s}}$. The decay of a fluid flow is a non-linear process. From the non-linear terms in the fluid equations (8.16) one can form the non-linearity timescale $\tau_{\mathrm{nl}} \sim L_{\mathrm{f}}/V_{\mathrm{f}} \sim R_*/\sqrt{K}$. The approximate lifetime of the gravitational wave source is then $\tau_{\mathrm{v}} = \min(\tau_{\mathrm{nl}}, H_{\mathrm{n}}^{-1})$.

---

[7]Recent work has demonstrated that a fifth parameter, the sound speed, should be added to the list [69, 94]. Holographic calculations indicate that at phase transitions in strongly coupled theories the sound speed can be quite different from $1/\sqrt{3}$ [95]

[8]While the four parameters listed above allow one to fit the gravitational wave spectra obtained from numerical simulations, it has been found that two of these parameters, the strength of the phase transition $\alpha$, and the inverse phase transition duration $\beta$, are not entirely independent from each other [72, 73]: Stronger phase transitions, characterized by larger values of $\alpha$, take more time to complete and hence also predict a smaller value of $\beta$. Conversely, weaker phase transitions, predicting smaller values of $\alpha$, proceed rather quickly. In simple words, the underlying reason for this relationship is rooted in the fact that the strength of the phase transition is related to the potential difference between the metastable and the stable ground state, $\Delta V$. As the temperature drops, the difference between the metastable and the stable ground state becomes larger. Strong phase transitions are a result of a sufficient amount of supercooling, i.e., the phase transition proceeds well below the critical temperature, where the two minima are degenerate, and hence take more time to complete.

A more careful calculation of gravitational wave production in an expanding universe [98] shows that a better approximation for the effective source lifetime is

$$\tau_{\rm v} = H_{\rm n}^{-1}\left(1 - \frac{1}{\sqrt{1 + 2\tau_{\rm nl}R_*}}\right), \tag{8.19}$$

which assumes that the RMS fluid velocity stays constant until $\tau_{\rm nl}$, when it immediately vanishes. Hence we can estimate

$$\Omega_{\rm gw} \sim \left(1 - \frac{K^{1/4}}{\sqrt{K^{1/2} + 2H_{\rm n}R_*}}\right)(H_{\rm n}R_*)K^2, \tag{8.20}$$

which goes as $(H_{\rm n}R_*)^2 K^{3/2}$ for fluid flow lifetimes much less than the Hubble time. We recall from Eq. (7.43) that the kinetic energy fraction is

$$K \simeq \frac{\kappa\alpha_{\rm n}}{1 + \alpha_{\rm n}}, \tag{8.21}$$

where $\kappa(\alpha_{\rm n}, v_{\rm w})$ is the efficiency with which Higgs potential energy is turned into kinetic energy, discussed around Eq. (7.44).

A more detailed survey of the expected lifetimes and correlation times of the various sources can be found in Ref. [12]. It is important to note that numerical simulations and modelling show that the dimensionless constant of proportionality in Eq. (8.20) is O($10^{-2}$) and not of order unity [71].

The shape of the gravitational wave spectrum is best understood for acoustic production, which has been explored by numerical simulations [71] and are now described by a precise theoretical framework, the sound shell model [66]. This is only the most recent development in a long history of modelling of gravitational wave production at a thermal phase transition [68, 99–102].

As sketched in Fig. 20 (left) the power rises as $k^9$ to a domed peak at $kR_* \sim 10$, and then decreases as $k^{-3}$. If the wall speed is close to the speed of sound, $v_{\rm w} \sim c_{\rm s}$, then the domed peak becomes a slow $k^1$ increase towards a peak wavenumber determined by the sound shell thickness $\Delta R_*$. At very low wavenumbers, where the signal is generically small, different power laws may emerge [103, 104]. For wall speeds away from the sound speed, the shape of the domed peak can be approximated by a simpler broken power law rising as $k^3$ and falling as $k^{-4}$ [71].

The turbulent stage, however, is far less well understood, as simulations of turbulent flows are very challenging [97, 107], and consequently the modelling [102, 105, 106] relies on untested assumptions. The main assumptions that go into the modelling are how the flow autocorrelation time depends on wavenumber, and also how the global quantities such as kinetic energy and correlation length evolve with time. The differences between these assumptions explain the different predictions for the power laws at low and high wavenumber, sketched in Fig. 20 (right). Only dedicated numerical simulations can resolve these issue. The simulations of turbulent vortical flow [107] seem to disagree with all the predictions, having a slow rise of around $k^1$ to the peak, and a $k^{-8/3}$ decay at high wavenumber.

Other unresolved questions are how the vortical component of the flow, which leads to turbulence, is generated in the first place, how important it is relative to the the compressional part, and how efficient they are at generating gravitational waves. On the first question, recent simulations [97] show that vorticity is generated by bubble collisions in strong phase transitions. For example, in the simulation shown in Fig. 21), the vorticity is associated with regions where bubbles collide, and about 20% of the fluid kinetic energy is in vortical flow. The figure also shows the above-mentioned heating of the symmetric phase, here confined to shrinking asymmetrical droplets.

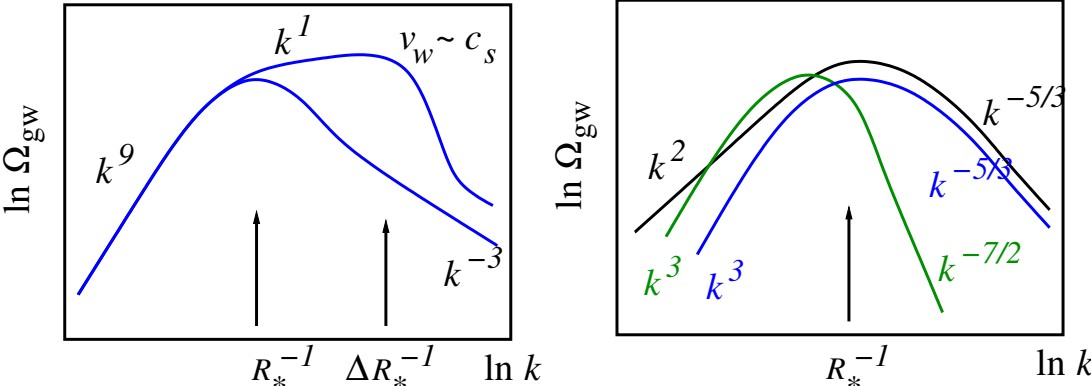

Figure 20: *Left panel:* shape of the gravitational wave spectrum from sound waves, showing the power laws and the characteristic wavenumber scales, which are determined by the mean bubble centre separation $R_*$ and the sound shell thickness $\Delta R_*$. *Right panel:* survey of predictions of the power spectrum from modelling of turbulent gravitational wave production, from [105] (green), [102] ("top-hat" autocorrelation, black), and [106] (blue).

There is clearly much work to be done before we are able to accurately compute a gravitational wave power spectrum for a phase transition of any strength. At the moment, this means regions where simulations have been carried out [71] and models [66] have reasonable agreement in the range $0.4 \lesssim v_{\mathrm{w}} \lesssim 0.9$ and $\alpha_{\mathrm{n}} \lesssim 0.1$.

For ultra-strong transitions ($\alpha_{\mathrm{n}} \gg 1$), the energy density of the universe becomes dominated by the potential energy of the scalar field, and the expansion of the universe starts to accelerate. This is the original idea behind cosmic inflation [4], and it was quickly realised that the nucleation rate parameter cannot be much smaller than the Hubble rate, as otherwise the majority of the Universe would still be inflating [65, 108]. Close to this boundary, the bubble separation $R_*$ would be of order the Hubble length, and so the gravitational signal would be maximal, motivating recent study [92, 109, 110]. In particular, bubble dynamics in the near-vacuum state is rather different from the thermal energy dominated case we consider in these lectures [92]. Recent numerical simulations involving just the scalar field are helping to develop a picture of what happens when the fluid plays no role at all [55, 93, 111]. One old idea which can be tested at the same time is that vacuum bubble collisions could have generated primordial black holes [108, 111–113].

The gravitational wave power spectrum at much larger scales than $R_*$ is also not fully understood. There are predicted to be some characteristic power laws in the gravitational wave power spectrum at long wavelengths connected with expanding shells of shear stress [103, 104, 114, 115], which in the strong supercooling case would not be dominated by the sound shell model signal. The fluid flow length scale is also expected to grow due to non-linearities in the fluid equations, which may also contribute to the large-scale power [116].

## 8.3 Comparison with GW observations

Finally for this section, we see how calculations of the gravitational wave signal match up to the potential for observations. First, we study how the gravitational wave frequency and intensity change between their generation and their observation.

The frequency scale of the spectrum when it is produced can be taken to be $f_* = R_*^{-1}$, which is also an estimate of the peak frequency. The frequency today $f_{*,0}$ is determined by the redshifting of the gravitational waves since the time of the phase transition. Using the Fried-

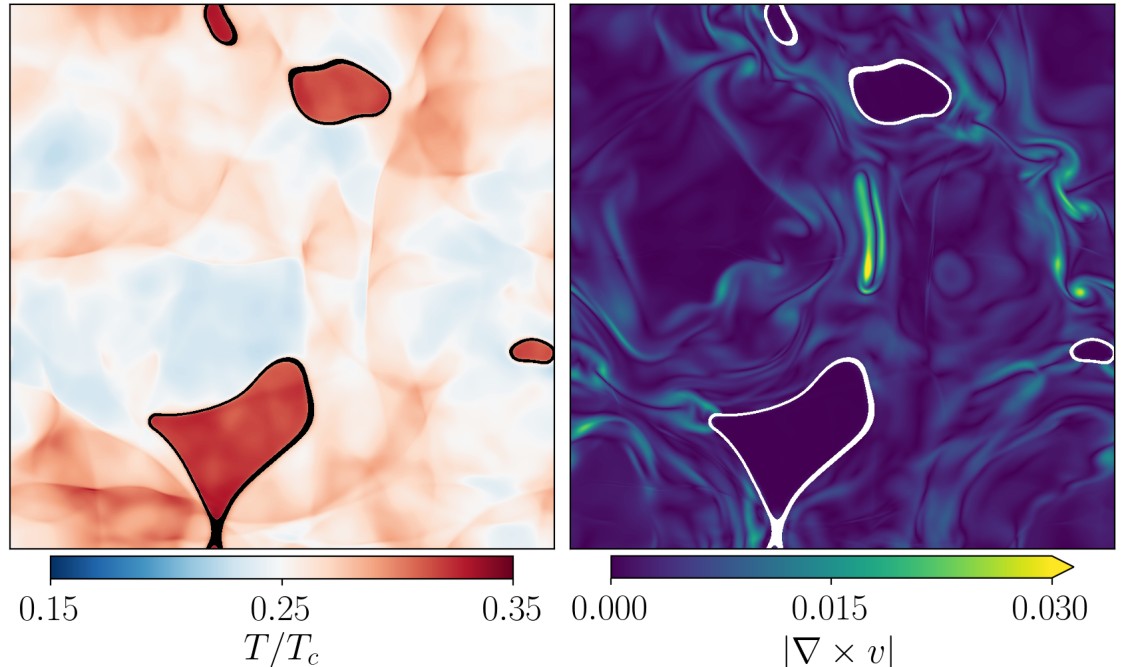

Figure 21: Slice through a numerical simulation of the field-fluid system, showing temperature (left) and vorticity (right) following a strong phase transition, with $\alpha_n = 0.5$ and $v_w = 0.44$ [97] . The temperature is highest inside shrinking droplets of the symmetric phase. The vorticity (shown in units of the critical temperature $T_c$) is highest around the sites of bubble collisions. (Figure courtesy D. Cutting.)

mann equation, the conservation of entropy, and a photon temperature today of $T_{\gamma,0} = 2.725$ K, one can show that

$$f_{*,0} \simeq 2.62 \left( \frac{1}{H_n R_*} \right) \left( \frac{T_n}{10^2 \, \text{GeV}} \right) \left( \frac{g_{\text{eff}}}{100} \right)^{\frac{1}{6}} \mu\text{Hz} . \tag{8.22}$$

Hence LISA will be most sensitive to electroweak-scale (100 – 1000 GeV) transitions with bubble separations between $10^{-2}$ and $10^{-3}$ of the Hubble length.

The intensity of the gravitational wave signal is also affected by the expansion of the Universe. While the energy density of the Universe is dominated by radiation, the density parameter of gravitational waves $\Omega_{\text{gw}}$ remains constant, as the energy densities of both components redshift the same way. However, once radiation gives way to non-relativistic matter as the dominant component, the density parameter of gravitational waves $\Omega_{\text{gw}}$ decreases, until today it is reduced by a factor

$$F_{\text{gw},0} = (3.57 \pm 0.05) \times 10^{-5} \left( \frac{100}{g_{\text{eff}}} \right)^{\frac{1}{3}} , \tag{8.23}$$

where the uncertainty comes mainly from the measurement of the Hubble rate today, taken here to be the Planck 2015 best-fit value $H_0 = 67.8 \pm 0.9 \, \text{km s}^{-1} \, \text{Mpc}^{-1}$ [117].

We can now present a simple model for the gravitational wave power spectrum from first-order phase transitions, which captures the parametric understanding of the power outlined in this section, gives a simple functional form based on numerical simulations [12, 71], and includes an explicit attenuation factor due to the decay of the flow [98]. The contribution to the fractional density in gravitational waves in a logarithmic frequency interval $\text{d} \ln f$ from a

phase transition with mean bubble centre separation to Hubble length ratio $H_n R_*$ generating a kinetic energy fraction $K$ is

$$\frac{d\Omega_{\text{gw,0}}}{d\ln f} = 2.061 F_{\text{gw,0}} \tilde{\Omega}_{\text{gw}} \frac{(H_* R_*)^2}{\sqrt{K} + H_* R_*} K^2 C(f/f_{\text{p,0}}), \tag{8.24}$$

where $\tilde{\Omega}_{\text{gw}} = 0.012$ is a numerically determined constant, representing the approximate efficiency with which kinetic energy is turned into gravitational waves, $C(s)$ is the function

$$C(s) = s^3 \left(\frac{7}{4 + 3s^2}\right)^{7/2}, \tag{8.25}$$

describing the gravitational wave power around the peak, and $f_{\text{p,0}} = 10 f_{*,0}$ is an estimate of the peak frequency, again from the numerical simulations. The number 2.061 is an approximation to $3/\int_0^\infty d\ln(s) C(s)$. In this very simple model the power at small $s$ is over-estimated, and at large $s$ under-estimated, compared to the sound shell model, and has significant deviations around the peak for wall speeds near the speed of sound. Its domain of reliability is $\alpha_n \lesssim 0.1$, $0.4 \lesssim v_w \lesssim 0.5$ and $v_{\text{CJ}}(\alpha_n) \lesssim 0.9$, where $v_{\text{CJ}}$ is the minimum speed of a detonation, given in Eq. (7.34). This represents the parameter range where gravitational wave power spectra have been extracted from numerical simulations. The model does not get the shape right for deflagrations with $v_w$ near or above the speed of sound (see Fig. 20).

We are now in a position to estimate the expected density fraction in gravitational waves from a phase transition, or equivalently the expected characteristic strain. As the power spectrum has a definite peak, the estimate represents both the total power and the peak power. Taking $T_n = 500$ GeV, $\alpha_n = 0.3$, $\beta/H_n = 10^2$, and $v_w = 0.4$, we find from Eqs. (8.22), (8.20), (8.21), and (7.44) that

$$\Omega_{\text{gw,0}} \sim 10^{-12}, \tag{8.26}$$

and that the peak frequency estimate is $f_{*,0} \simeq 10^{-3}$ Hz. At this frequency, the characteristic strain of a signal with a gravitational wave density parameter of order $10^{-12}$ is

$$h_c \sim 10^{-21}, \tag{8.27}$$

where we have used Eq. (8.10). Estimates of the power spectrum, in the simplified broken power law approximation in Eq. (8.24), can be generated by the PTPlot tool [12, 118].

In Fig. 19 we have plotted the signal from a phase transition predicted by the more refined sound shell model [66, 119], with the parameters given above. While the strength of the transition is in the range where the amplitude and shape of the signal are merely indicative, the amplitude agrees with our order-of-magnitude estimate above, and such a transition would be detected very clearly by LISA, despite the competing astrophysical signals. The most significant of these will probably be the foreground from white dwarf binaries in our galaxy, of which there are thought to be tens of millions. This foreground will always be present, unlike the transient signals from merging supermassive black holes in distant galaxies. However, the anisotropy of the galactic white dwarf foreground should make it vary in strength throughout the year as LISA orbits the sun and the relative orientation of the plane of the satellite constellation and the galaxy changes. This will make it distinguishable from the isotropic background expected from phase transitions (see [83] for a review of gravitational wave signals and detection methods).

## 9 Summary

Gravitational waves are an extraordinary new tool for astronomy and cosmology, and can be used to directly observe processes which happened in the early universe. In particular, first-order phase transitions in the early universe give rise to gravitational waves with characteristic

power spectra. The source of the gravitational waves is the shear stresses in the fluid set up by the expansion and collision of bubbles of the low-temperature symmetry-broken phase during the phase transition. The shear stresses are initially overlapping pressure waves, or sound, which may be strong enough to generate turbulence when they collide.

The dynamics of the phase transition can be described in terms of only four properties of the phase transition: the bubble nucleation temperature $T_n$, the strength parameter $\alpha_n$, the transition rate parameter $\beta$, and the bubble wall speed $v_w$. These are all in principle computable from an underlying particle physics model, and in these lectures we have gained some insight into the methods. We have also seen how relativistic combustion theory explains how kinetic energy is generated in the fluid, which eventually shows up as the sound waves and turbulent motion.

The current excitement is that information about these phase transition parameters is contained in the power spectrum of the gravitational waves, and that the space mission LISA could detect these gravitational waves if the phase transition took place at a time around $t = 10^{-11}$s [12], when the temperature was in the range 100 GeV to 1 TeV. This is the scale of electroweak symmetry-breaking. Hence LISA is a particle physics experiment, as well as an astrophysical observatory. A first-order electroweak phase transition is possible only if there is physics beyond the Standard Model, perhaps in the form of extra Higgs fields. LISA has the potential to discover new fundamental physics.

This is a dynamic and developing field. In the last section we already mentioned the current uncertainty about the hydrodynamics of strong phase transitions ($\alpha_n \gtrsim 0.1$). It is also not yet known how accurately LISA can measure the phase transition parameters, but even a 20% measurement of (say) the wall speed is not yet matched by the accuracy of the calculations from underlying particle physics models. Furthermore, the framework of homogeneous nucleation theory is taken from condensed matter, where nucleation by thermal fluctuation has never been observed to take place. There is a long-standing mystery about the lifetime of the metastable superfluid $^3$He-A phase towards decay into $^3$He-B. The theory described in these lectures predicts lifetimes of experimental samples of superfluid $^3$He-A of order $10^{1000000}$ seconds, while in the laboratory, even when taking great care over impurities, $^3$He-A lasts only a few minutes (see e.g. [120]). There is much to be done to realise the goal of making LISA into a particle physics experiment to complement the Large Hadron Collider.

## Acknowledgements

These notes are based on a course given by Mark Hindmarsh at the 24th W.E. Heraeus Summer School "Saalburg" for Graduate Students on "Foundations and New Methods in Theoretical Physics" in September 2018. We would like to thank the organisers and all participants for the warm and productive atmosphere. MH would also like to thank the organisers of the 2017 Benasque School for Gravitational Waves for Cosmology and Astrophysics, where an earlier version of these lectures were given.

We are grateful to Fëanor Reuben Ares, Giordano Cintia and Anupam Mazumdar for careful reading and useful comments on the manuscript. We further thank Daniel Cutting for Figs. 9, 14, 21 and Chloe Gowling for scripts used in the production of Fig. 19.

**Funding information**   MH acknowledges support from the Science and Technology Facilities Council (grant number ST/L000504/1) and the Academy of Finland (project numbers 286769, 333609). MP acknowledges support by the Heidelberg Graduate School of Fundamental Physics and through a scholarship of the German Academic Scholarship Foundation. JL acknowledges support by an Emmy-Noether grant of the DFG under grant number Ei/1037-

1. ML is supported by a PhD grant from the Max Planck Society.

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
