# Peer review of "Phase transitions in the early universe"

_SciPost Physics Lecture Notes, doi:SciPost Phys. Lect. Notes 24 (2021)_

## Round 2 · Referee Report · Anonymous (Referee 1) · 2020-11-2

Strengths

Excellent review of the subject, complete and pedagogical.

Weaknesses

The spectral shape of the GW signal is still work in progress, but this is not a weakness of the lecture notes, it simply reflects the state of the art in the field.

Report

I strongly recommend publication of these lecture notes, which collect in a unique piece of work a relevant amount of scattered information, rendering them very useful both for the community and for students wishing to learn about phase transitions in the early universe.

---

## Round 2 · Referee Report · Anonymous (Referee 2) · 2020-11-3

Strengths

The review is well written; relevant topics are discussed in some details. I believe it will be a good addition to several reviews written in this popular research direction which will keep the scientific community busy for a long time.

  • I have taken a stance that this is a scientific discussion, and I will be happy to present myself as your referee. I am Anupam Mazumdar, one of the authors of your Ref.[17].

Weaknesses

I do not see any obvious weakness in the review.
The subject is topical, and despite several reviews written in this field of research, this review is a welcoming addition.

Report

I have a few requests to the authors. They may wish to add the discussion at their discretion. These suggestions will only help the review further, in my opinion.

a) A broader view on phase transition would be constructive in the introduction. As the authors know very well that one can have both thermal/non-thermal phase transitions in the early Universe, and either of them can trigger gravitational waves; via preheating, or during fragmentation of the inflation condensate, see your Ref.[17] and references therein. This will make the discussion more rounded. Otherwise, the reader may be forced to take the message that there is only one type of phase transition, i.e., thermal.

b) Slightly technical issue, but I trust the authors can do a perfect job here; it is related to the fact that the loop corrections to the effective potential at either zero temperature or in a finite temperature are not real everywhere in the field space. This has been discussed by several notable authors,
see https://arxiv.org/abs/1510.07613, https://www.sciencedirect.com/science/article/abs/pii/0370269388908623?via%3Dihub,
https://journals.aps.org/prd/abstract/10.1103/PhysRevD.36.2474
It would be very nice to discuss this topic here as well. Furthermore, it will help the readers a lot.

c) Fig. [20] is very instructive, but Fig. [21] does not add much physics. It gives the impression that the ball is sliding away on a dusty pitch! it may be worth pausing and explaining the physics bit better. A simulation can only be taken with a grain of salt, so worth giving the assumptions behind these simulations.

d) Fig.[20] is very nice, but what are the other competitors in the market; for instance, astrophysics sources, have we exhausted all the spectra of the astrophysical sources from redshift, z=8-9 - to- all the way to 3-4 where LISA would be sensitive to? I am not an expert in this field myself, but introducing a paragraph won't hurt the readers.

Of course, these are minor modifications, and it is not necessary as such, except it will improve your article a notch better that already extremely well-presented review in my opinion.
  • validity: good
  • significance: good
  • originality: ok
  • clarity: good
  • formatting: good
  • grammar: good

Author:  Martin Pauly  on 2021-01-06  [id 1126]

(in reply to Report 2 on 2020-11-03)

We thank the referee for their time and comments on the manuscript.
In the following we briefly address the helpful comments in the report:

a) We have added a paragraph on non-thermal phenomena in the introduction that also points interested readers to further references.

b) We have decided to not discuss this topic in detail, as it would render our discussion of the thermal potential a lot more technical. Instead, we have added a paragraph with relevant references that go into more depth on this topic on page 10.

c) We have added a few sentences that go into more detail on the physics of Fig. 21 - we hope that this makes the physics behind that figure clearer.

d) We have added a comment on this in Sec. 8.1 that points out that there are also various astrophysical sources. For more details we have added relevant references.

We thank the referee for pointing out these possibilities for improvement.

---

## Round 3 · List of Changes

We have addressed the helpful comments in the report of Referee 2, see the response to that report for more detail.

We have also corrected typos and added citations to other planned space-based detectors at the end of section 8.1. Further, we have extended the discussion of the fitting parameters by adding the new footnote 8 in section 8.2.

---

## Editorial Decision

published